# SIV-induced terminally differentiated adaptive NK cells in lymph nodes associated with enhanced MHC-E restricted activity

Nicolas Huot [1], Philippe Rascle[1,2], Caroline Petitdemange[1], Vanessa Contreras[3], Christina M. Stürzel[4], Eduard Baquero[5], Justin L. Harper [6], Caroline Passaes [1], Rachel Legendre [7], Hugo Varet [8], Yoann Madec [9], Ulrike Sauermann [10], Christiane Stahl-Hennig[10], Jacob Nattermann[11], Asier Saez-Cirion [1], Roger Le Grand[3], R. Keith Reeves [12], Mirko Paiardini [6,13], Frank Kirchhoff [4], Beatrice Jacquelin [1] & Michaela Müller-Trutwin [1✉]

Natural killer (NK) cells play a critical understudied role during HIV infection in tissues. In a natural host of SIV, the African green monkey (AGM), NK cells mediate a strong control of SIVagm infection in secondary lymphoid tissues. We demonstrate that SIVagm infection induces the expansion of terminally differentiated NKG2a$^{low}$ NK cells in secondary lymphoid organs displaying an adaptive transcriptional profile and increased MHC-E-restricted cytotoxicity in response to SIV Env peptides while expressing little IFN-γ. Such NK cell differentiation was lacking in SIVmac-infected macaques. Adaptive NK cells displayed no increased *NKG2C* expression. This study reveals a previously unknown profile of NK cell adaptation to a viral infection, thus accelerating strategies toward NK-cell directed therapies and viral control in tissues.

[1] Institut Pasteur, Unité HIV, Inflammation et Persistance, Paris, France. [2] Université Paris Diderot, Sorbonne Paris Cité, Paris, France. [3] CEA-Université Paris Sud-Inserm, U1184, IDMIT Department, IBFJ, Fontenay-aux-Roses, France. [4] Ulm University Medical Center, Ulm, Germany. [5] Institut Pasteur, Unité de Virologie Structurale, Paris, France. [6] Division of Microbiology and Immunology, Yerkes National Primate Research Center, Emory University, Atlanta, GA, USA. [7] Bioinformatics and Biostatistics Hub, Department of Computational Biology, Institut Pasteur, Paris, France. [8] Biomics Platform, Center for Technological Resources and Research (C2RT), Institut Pasteur, Paris, France. [9] Institut Pasteur; Epidemiology of Emerging Diseases Unit, Paris, France. [10] Deutsches Primatenzentrum - Leibniz Institut für Primatenforschung, Göttingen, Germany. [11] Medizinische Klinik und Poliklinik I, Universitätsklinikum Bonn, Germany; German Center for Infection Research (DZIF), Bonn, Germany. [12] Center for Virology and Vaccine Research, Beth Israel Deaconess Medical Center, Harvard Medical School, Boston, MA, USA. [13] Department of Pathology and Laboratory Medicine, Emory University School of Medicine, Atlanta, GA, USA. ✉email: mmuller@pasteur.fr

The impact of NK cells on viremia in HIV-1 infection has been clearly demonstrated through studies on HLA-I alleles[1–3]. Burgeoning data show that the NK cell repertoire is not stable and that its diversity increases with age[4–6]. The NK cell repertoire diversity reflects immune history and correlates with viral susceptibility[7]. This raises the possibility that there are unexpected functional specialization and distinct adaptive capabilities among NK cell subpopulations in response to viral infections and immunizations, a mechanism which could be exploited to promote antiviral immunity[8–12]. NK cells have been reported to differentiate not only in thymus and bone marrow but also within liver, uterus, and secondary lymphoid tissues (SLT). However, data on NK cell differentiation in SLT during a viral infection are scarce[13–16]. African non-human primates, such as African green monkeys (AGM), have been a natural host for simian immunodeficiency viruses (SIV) for probably over 1 million of years[16–18]. In contrast to people living with HIV (PLH), natural SIV hosts typically do not progress toward disease despite persistent high viremia[19]. We have previously shown that AGMs efficiently control viral replication in SLT (both in the T zone and in B cell follicles), and that this viral control is mediated predominantly by NK cells[20]. No such viral control is generally observed in lymph nodes (LN) of PLH and in the non-human primate model of HIV, i.e., SIVmac-infected macaques (MAC), where SLT constitute a major viral reservoir[21,22]. The capacity to control viral replication in B cell follicles of AGM was associated with the presence of CXCR5[+] NK cells[20].

Here, we hypothesize that non-pathogenic (AGM) and pathogenic HIV-1/SIVmac infections differentially imprint NK cells and thereby contribute to the the distinct capacity of the NK cells to control SIV/HIV replication in SLT. In this work, we show that SIVagm infection induces the expansion of terminally differentiated NKG2a[low] NK cells in SLT displaying an adaptive transcriptional profile and increased MHC-E-restricted cytotoxicity in response to SIV Env peptides. THEMIS but not NKG2C expression was increased in the adaptive NK cells. Our longitudinal analysis also uncovers, that such NK cell terminal differentiation is blocked in SIVmac-infected macaques, in which NK cells showed a decrease in MHC-E-restricted cytotoxicity in response to SIV Env peptides and rather frequently expressed IFN-γ. This study reveals a distinct NK cell adaptation to a viral infection. It also improves our understanding of NK cell dysfunction in HIV infection and thereby opens avenues to improve NK-cell mediated viral control in HIV cure strategies.

## Results

**Blood NK cell diversity during SIV infection in the natural versus heterologous host.** NK cell effector function is known to increase together with the diversity of the receptor repertoire on NK cells[23–26]. We first estimated and compared phenotypic changes in blood NK cells following SIV infection between pathogenic and natural host models. The NK cell gating strategy and animals are describe in, respectively, Supplementary Figs. 1 and 2. Force-directed clustering analysis was performed based on expression data from nine markers of NK cells followed longitudinally (between day 0 and day 240 p.i.) in 6 MAC and 6 AGM. This generated NK cell cluster profiles that were different between AGM and MAC following SIV infection but consistent between monkeys from a same species (Fig. 1a, b and Supplementary Fig. 3a, b). Both in AGM and MAC, IFN-γ and perforin showed the strongest positive correlation to each other (Supplementary Fig 3c, d). However, only in AGM, markers studied longitudinally throughout infection were often inter-correlated, while less or not correlated in MAC (Supplementary Fig. 3c, d). For instance, in SIVagm infection, IFN-γ correlated with the

expression of the activating receptors Nkp46, NKp30, and NKG2D, but this was not the case during SIVmac infection. These results suggest a more coordinated action of NK cells during SIVagm than SIVmac infection. When we analyzed the association between viremia levels and the receptor repertoire, the only negative correlation observed was between viremia and CD16 in AGM ($p = 0.006$, $\rho = -0.3$; Supplementary Fig. 3c, d).

We next analyzed distinct NK cell subpopulations stratified by CD16 and NKG2$_{a/c}$, two markers commonly used to define NK cell terminal maturation in humans (Supplementary Fig. 1). Follow up of these subsets during infection in blood did not reveal major differences within each species studied, but highlighted profound differences between AGM and MAC, such as an expansion of NKG2$_{a/c}$[high]CD16[-] NK cells in SIVmac infection (Fig. 1c, d). We then analyzed the functional activity of these blood NK cell subsets based on the ex vivo expression of the CD107a surrogate marker. The activity decreased in all subsets in response to SIV infection for both AGM and MAC (Supplementary Fig. 3e, f). However, the activity of NKG2$_{a/c}$[low]CD16[+/−] NK cells returned rapidly to normal levels in AGM but not in MAC.

In order to evaluate the differentiation and functionality of the NK cells during SIV infection in a comprehensive way, we implemented a genome-wide transcriptional mapping. We sorted the four NK cell subpopulations stratified by CD16 and NKG2$_{a/c}$ expression from the blood of chronically infected monkeys. We identified 2586 and 1895 genes to be differentially regulated between the distinct NK cell subpopulations in MAC and AGM, respectively (Fig. 2a). We analyzed in more detail the gene expression profiles of NK cell receptors (Fig. 2b), transcription factors (Fig. 2c), and effector molecules (Fig. 2d). Genes upregulated in NKG2$_{a/c}$[high]CD16[−] NK cells included those associated with tissue homing (CXCR3), proliferation (MKI67), and transcription factors involved in cellular stemness and quiescence such as Tcf7 (TCF-1), indicating a low differentiation stage when compared to the other subsets[27,28]. In contrast, NKG2$_{a/c}$[low]CD16[+] NK cells expressed elevated levels of genes associated with tissue egress and circulation (CX3CR1) and genes encoding transcription factors associated with effector function and terminal differentiation, including TBX21 (T-bet) and ZEB2[29,30]. The other two NK cell subpopulations displayed gene expression profiles of intermediate differentiation. Thus, the combination of NKG2$_{a/c}$ and CD16 allowed the determination of distinct NK cell differentiation stages in non-human primates (NHP). Together these results uncovered profound differences in NK cell behavior following SIV infection between the two models that seem to be driven by differences in the terminal differentiation process.

**NK cell terminal maturation and functional activity in lymph nodes is driven differentially between SIVagm and SIVmac infection.** Critical steps in NK cell development including terminal differentiation necessitate distinct microenvironments within extramedullary tissues[4,31]. The latter include the liver and gravid uterus as well as SLT such as LN, where we previously identified a strong capacity to control SIV replication in AGM in contrast to MAC[20]. We evaluated the levels of natural cytotoxicity receptors (NCR), including among others NKp30, NKp46, NKG2D, and effector molecules before and until day 150 p.i in peripheral LN. The SPICE analysis showed, that the NK cell diversity increased in peripheral LN in both SIVmac and SIVagm infection (Fig. 3a). We evaluated if the NK cell terminal differentiation profiles in LN during SIV infection were different or not in the two models. The expression levels of NKG2$_{a/c}$ increased on LN NK cells and the NKG2$_{a/c}$[high]CD16[+] NK cells increased after SIVmac infection but not after SIVagm

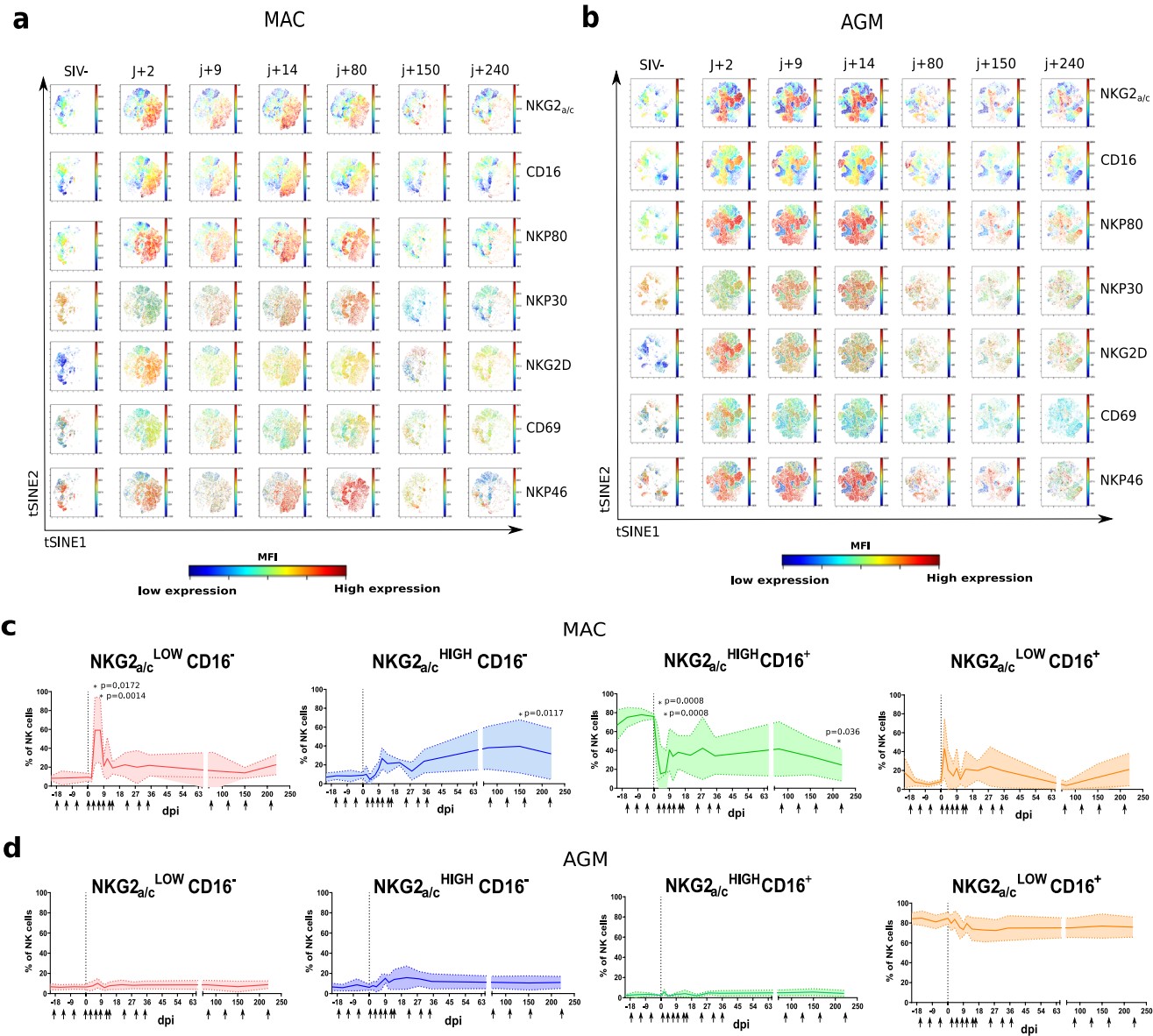

**Fig. 1 Longitudinal follow up of blood NK cells during SIV infection in the natural and heterologous hosts.** Six cynomolgus macaques (MAC) and six African Green Monkeys (AGM) were followed before and after SIV infection (acute and chronic infection up to day 250 p.i.). **a, b** viSNE analysis (Cytobank) on the concatenated flow cytometry data of **a** six MAC and **b** six AGM. Cells were stained with 12 markers and measured by flow cytometry. viSNE analysis was run on 80000 live CD45[+] CD3[-]CD20[-]CD14- single cells per sample using all surface markers. Data show expression on all viable single cells from blood, subjected to the tSNE algorithm, at the given time point. The red and blue colors indicate higher and lower mean fluorescence intensity (MFI) of a given marker. **c, d** Frequency of NK cell subpopulations over time in blood from **c** six MAC and **d** six AGM. The arrows indicate the analyzed time points. The full line represents the median, and the interquartile range is indicated by the area between the dotted lines. The color code used to define each of the four subpopulations is the same throughout the manuscript as described in Supplementary Fig. 1. For group comparisons two-sided Wilcoxon signed-rank test with Bonferroni correction were used ($n = 13$). P values of less or equal to 0.05 were considered statistically significant. Asterix indicate significant change when compare to the base line. Exact P value are provide on the graphs.

infection (Fig. 2b and Supplementary Fig. 4a, b). In SIVagm infection, the two NKG2$_{a/c}$$^{low}$ subsets (NKG2$_{a/c}$$^{low}$CD16$^+$ and NKG2$_{a/c}$$^{low}$CD16$^-$) increased in acute and chronic infection, respectively (Fig. 3b). These expansions were probably not a consequence of proliferation as the Ki-67 expression levels did not increase in the NKG2$_{a/c}$$^{low}$ cells (Fig. 3c). The NKG2$_{a/c}$$^{low}$ NK cells exhibited increased cytolytic activity (CD107a, Perforin) during primary and chronic SIVagm infection, but not in SIVmac infection, except for a transient response between days 2 and 14 p.i. (Fig. 3c and Supplementary Fig. 4e, f). NK cells from MAC showed ex vivo increases of IFN-ɣ expression in response to SIVmac infection, while in AGM, the expression

level of IFN-ɣ decreased in LN NK cells (Fig. 3c). AGM rather displayed increased expression of Perforin during primary SIVagm infection in contrast to SIVmac infection (Fig. 3c and Supplementary Fig. 4a, b). A correlation matrix showed indeed a positive correlation of NKG2$_{a/c}$ with CD16, IFN-ɣ, and NKG2D for MAC (Supplementary Fig. 4c), while AGM showed a negative correlation between NKG2$_{a/c}$ and perforin (PRF) (Supplementary Fig. 4d). In addition, more correlations were noticed again for NK cells in SIVagm as compared to SIVmac infection, suggesting, as in blood, a better coordination of NK cell response in SIVagm than SIVmac infection. Altogether, LN NK cells displayed higher expressions of NKG2$_{a/c}$ with a more

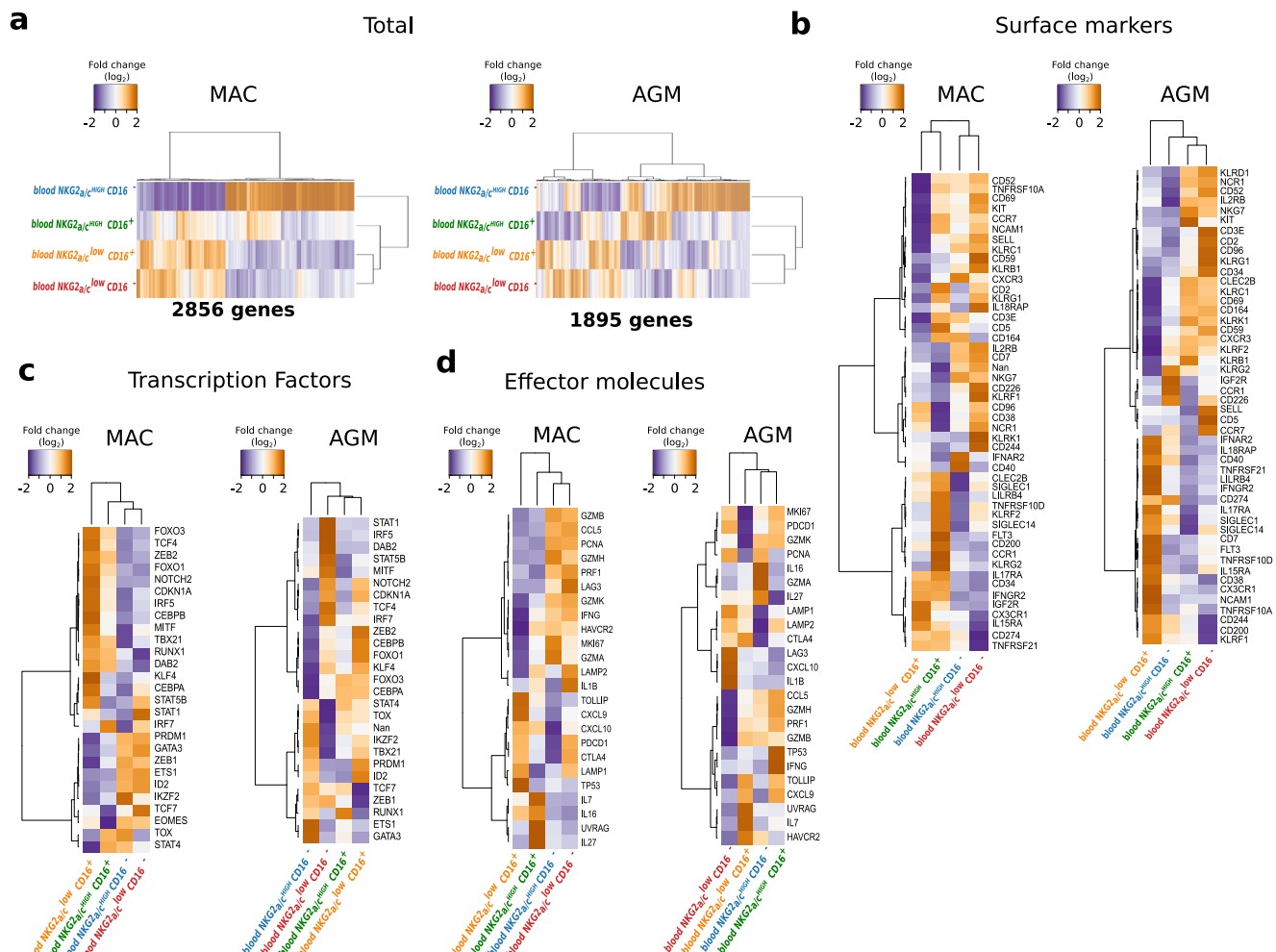

**Fig. 2 Genome-wide transcriptional and functional analysis of blood NK cells during SIV infection in the natural and heterologous hosts. a–d** Heatmaps of the RNA-seq transcript signatures for **a** total, **b** NK cell surface markers, **c** transcription factors, and **d** effector molecules in blood. The orange and blue colors indicate higher and lower levels of transcripts measured in NK cell subsets from three chronically infected monkeys per species. Each row represents a transcript showing a significant change, and each column represents the median of three chronically infected monkeys. In each panel, NK cell subsets are organized based on the overall similarity in gene expression patterns by an unsupervised hierarchical clustering algorithm of variable genes. A dendrogram, in which the length of the branches reflects the comparative difference in gene expression profiles between each of the NK samples is shown. A p-value adjustment was performed to account for multiple comparisons and control the false positive rate to a chosen level. P values ≤ 0.05 were considered statistically significant. The transcriptome similarity among clusters of the blood sample was evaluated by the Euclidean distance. In panel **b–d** the name of the proteins coding by the transcripts are indicated. Source data been deposited in the Gene Expression Omnibus database; the accession number is GSE140600.

pronounced cytokine (IFN-ɤ) profile in SIVmac infection, while a more cytotoxic profile (Perforin) correlating with low NKG2$_{a/c}$ expression was observed in SIVagm infection.

We analyzed if there was a relation between the observed NK cell subsets and viral load in LN. In MAC, only weak positive correlations were observed, i.e. between the NKG2$_{a/c}$high CD16+ NK cells and both cell-associated viral (ca-v) RNA (viral load in LN ($p = 0.02$) and ca-vDNA ($p = 0.048$) (Supplementary Table S1). In AGM, a positive correlation was observed for NKG2$_{a/c}$low CD16+ NK cells ($p < 0.008$ for ca-vRNA and ca-vDNA), while a negative correlation was observed with the NKG2$_{a/c}$high CD16− NK cells ($p < 0.03$ for ca-vRNA and ca-vDNA; Supplementary Table S1). No significant negative correlations were measured between NK cell markers and viral load in LN in MAC or AGM (Fig. 3d, e). The data suggest that viral replication is driving the expansion of NKG2$_{a/c}$high CD16+ NK cells in SIVmac infection but of NKG2$_{a/c}$low CD16+ NK cells in SIVagm infection.

In order to perform an unbiased analysis of NK cell subsets in SLT, we performed a genome-wide transcriptional mapping of the LN NK cells. Four NK cell subsets were sorted from the LN of the same animals and same time points as described above for the blood. The four subsets corresponded to NKG2$_{a/c}$low CD16−, NKG2$_{a/c}$high CD16−, NKG2$_{a/c}$low CD16+, and CXCR5+ NK cells. The latter were known to play a role on viral replication in SLT[20]. The NKG2$_{a/c}$high CD16+ subset was not sorted due to the extremely low number of these cells in LN. Heatmaps were drawn for genes coding for NK cell markers (Supplementary Fig. 5a), transcription factors (Supplementary Fig. 5b), and receptors associated with trafficking and immune function (Supplementary Fig. 5c). The CXCR5+ NK cells clustered separately from the other NK cell subpopulations in LN, but consistently closely to the NKG2$_{a/c}$low CD16- NK cells (Fig. 4a and Supplementary Fig. 5a–c).

The genome-wide analysis of all subsets in LN showed, in line with the observations made in blood (Fig. 2 and Supplementary

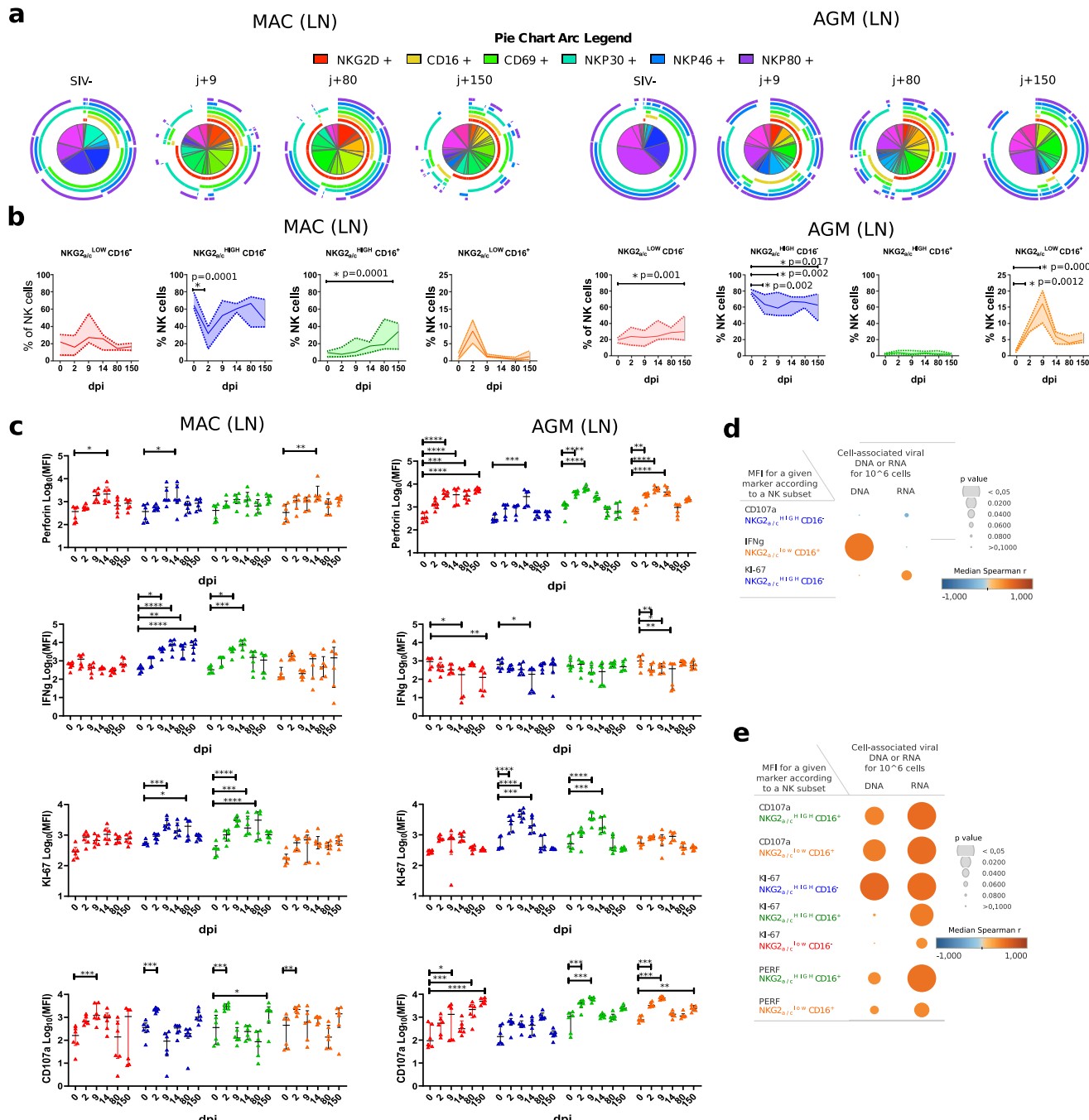

**Fig. 3 Dynamics of NK cell repertoire and activity in response to SIV infection in LN. a** Pie charts and arcs showing the proportion of different combinations of markers on NKG2a + NK cells isolated from LN during SIVmac (left panel) and SIVagm (right panel) infections, as analyzed using SPICE 5.3. **b** Frequency of NK cell subpopulations out of total NK cells in LN. Colors are the same as in Fig. 1. Cells were sampled before and during SIVmac and SIVagm infection (*n* = 6 animals per species) at the time point indicated by the *X* axis. The full line indicates the median, and the interquartile range is indicated by the area between the dotted lines. For Groups comparisons two-sided Wilcoxon signed-rank test with Bonferroni correction were used (*n* = 6). *P* values ≤ 0.05 were considered statistically significant. Asterix indicate significant change when compared to the base line. Exact *P* values are provided on the graphs. **c** Longitudinal profiles of intracellular levels of Perforin, interferon gamma, Ki-67, and surface level of CD107a on NK cell subsets ex vivo during SIVmac (left panel) and SIVagm (right panel) infections without any prior stimulation. The colors for each NK cell subpopulation are the same as in Fig. 1. Each symbol represents one monkey, the black line indicates the median and error bars the interquartile range. For group comparisons, two-sided Wilcoxon signed-rank test with Bonferroni correction was used (*n* = 6). *P* values ≤ 0.05 were considered statistically significant. Asterix indicate significant change when compared to the base line, graphically annotated as follows: \**p* < 0.05; \*\**p* < 0.01; \*\*\**p* < 0.001; \*\*\*\**p* < 0.0001. **d**, **e** Pearson correlation matrix between viral load in LN (ca-vDNA and ca-vRNA) and expression level of functional markers showed in Fig. 3c for **d** MAC and **e** AGM. The orange and blue colors indicate higher and lower *r* value. The *p* value is indicated by the size of the circle. All *p* and *r* values are given in the Data Source file.

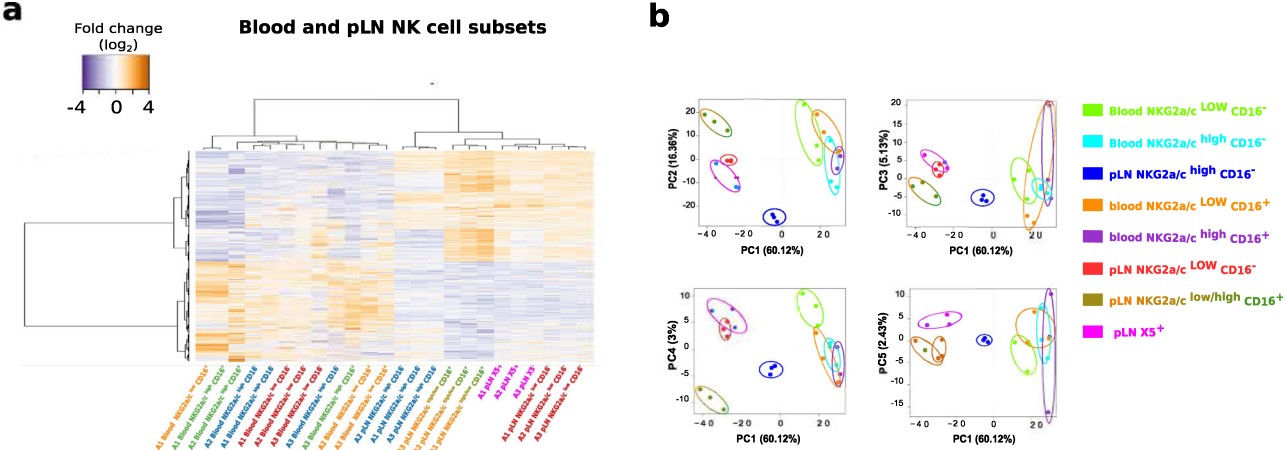

**Fig. 4 Deep transcriptional differences between blood and pLN NK cells subsets during chronic SIVagm infection. a** Heatmap showing the 5,489 genes differentially regulated between blood and LN NK cells in chronically SIVagm-infected AGM. The orange and blue colors indicate higher and lower levels of transcripts measured in blood and LN NK cell subsets from three chronically infected monkeys Each row represents a variable gene among clusters, and each column represents the NK cell subset for an individual monkey. The NK cell subsets are indicated in distinct colors. NK cell subsets are organized based on the overall similarity in gene expression patterns by an unsupervised hierarchical clustering algorithm of variable genes. A dendrogram, in which the length of the branches reflects the comparative difference in gene expression profiles between each of the NK samples is shown. A *p*-value adjustment was performed to account for multiple comparisons and control the false positive rate to a chosen level. The transcriptome similarity among clusters of the blood sample was evaluated by the Euclidean distance. *P* values ≤ 0.05 were considered statistically significant. Source data been deposited in the Gene Expression Omnibus database; the accession number is GSE140600. **b** Clustering of samples using the first five principal components (PC) of the indicated subset with percentages of variance associated with each axis based on the mean expression of the genes with variable expression. Each dot represents one monkey and circle around the dot represents one NK subset.

Fig. 5a–c), that the LN NKG2$_{a/c}$lowCD16$^+$ NK cells expressed high levels of markers known to be associated with NK cell terminal differentiation. Also, the NKG2$_{a/c}$highCD16$^-$ NK cells showed again the less differentiated profile and expressed high levels of transcripts coding for NKG2D (*KLRK1*), NKp46 (*NCR1*), and CD56 *(NCAM-1)* similar to the CD56$^{high}$ NK cell population observed in human SLT[15]. Differential expression of NK cell markers at the transcription level were confirmed at the protein level by flow cytometry (Supplementary Fig. 5d). This analysis confirmed that CD16 combined with NKG2$_{a/c}$ allowed distinguishing NK cell differentiation stages in LN. Taken together, this demonstrates that SIVmac infection led to an accumulation of only the intermediate, stage of NK cell differentiation in LN, whereas in SIVagm infection, the fully differentiated, cytolytic NKG2$_{a/c}$lowCD16$^+$ NK cells expanded.

**LN-specific NK cell gene signature during SIVagm infection in the natural host.** We next hypothesized that NK cells in LN, which are known to be able to eliminate SIVagm-infected cells in AGM[20], display LN-specific features distinct from NK cells in blood where the virus is not controlled. We therefore compared the genome-wide transcriptome profiles between blood and LN (Fig. 4a, b). Surprisingly, the CD16$^-$ and CD16$^+$ NK cells from LN were more closely related to each other than to their respective subsets in the blood. As many as 5489 genes displayed expression differences between LN and blood NK cells (Fig. 4a). The four NK cell subpopulations in the LN segregated clearly into distinct clusters, while they were less separated from each other in the blood (Fig. 4a,b). Gene Ontology (GO) enrichment analysis indicated that in LN from SIVagm-infected animals, NK cells displayed an enrichment for pathways involved in viral responses, regulation of defense, IFN response, pattern recognition receptor signaling and inflammasome activation (NLRP3, NLRp1, and NLRC4) compared to blood NK cells (Supplementary Fig. 6a–c). In contrast, blood NK cells displayed an enrichment in biological process terms,

such as regulation of stress-activated mitogen-activated protein kinase (MAPK) cascade and of histone modifications (Supplementary Fig. 6d–f). Thus, LN NK cells displayed a common gene signature distinct from blood during SIVagm infection in the natural host showing for instance stronger viral defense and inflammasome activation.

**Adaptive NKG2$_{a/c}$lowCD16$^+$ NK cells in LN during SIVagm infection.** We next addressed the question of the overall anti-viral activity profiles in the NK cells which expanded in LN upon acute (CXCR5$^+$, NKG2$_{a/c}$lowCD16$^+$) and/or chronic SIVagm infection (CXCR5$^+$, NKG2$_{a/c}$lowCD16$^-$) using the genome-wide expression data (Fig. 5). We compared these three NK cell subsets to the remaining subset (NKG2$_{a/c}$highCD16$^-$). The NKG2$_{a/c}$lowCD16$^+$ NK cells were those with the highest number of upregulated genes (1072 genes) compared to NKG2$_{a/c}$highCD16$^-$ NK cells followed by 432 upregulated genes in NKG2$_{a/c}$lowCD16$^-$ and 294 genes in CXCR5$^+$ NK cells. As many as 176 genes were commonly upregulated among the three expanded subsets, and one gene was downregulated, *CCDC102B* (Fig. 5a). Among the 176 upregulated genes, GO enrichment analysis uncovered pathways important for the lytic activity of NK cells, such as pathways associated with the biology of cytotoxic granules, including lytic vacuoles, secretion by cells, lysosomes and lytic vacuole membranes, cellular assembly and microtubule cytoskeleton (Fig. 5b, c). Among the genes related to cytokines were *CXCL11, CXCL12, IL-27, IFNGR, IL15RA,* and *IL21R* and inflammasome-related proteins included *NLRP3* and *IL1B* (Fig. 5c). Many of the other upregulated genes are known to be associated with metabolism, such as *GLUL* (glutamine synthetase); H6PD (Glucose-6-phosphate-dehydrogenase), *ENTPD1* (CD39), and *ATP13A2*. The NKG2$_{a/c}$low NK cells also expressed high levels of *Lamp1* (CD107a; *p* < 0.000000005) and *CCL5* (*p* = 0.0003; Fig. 5c). These data are consistent for CXCR5$^+$ and NKG2$_{a/c}$low NK cells, and in particular for NKG2$_{a/c}$lowCD16$^+$ NK cells, displaying a strong degranulation activity in LN.

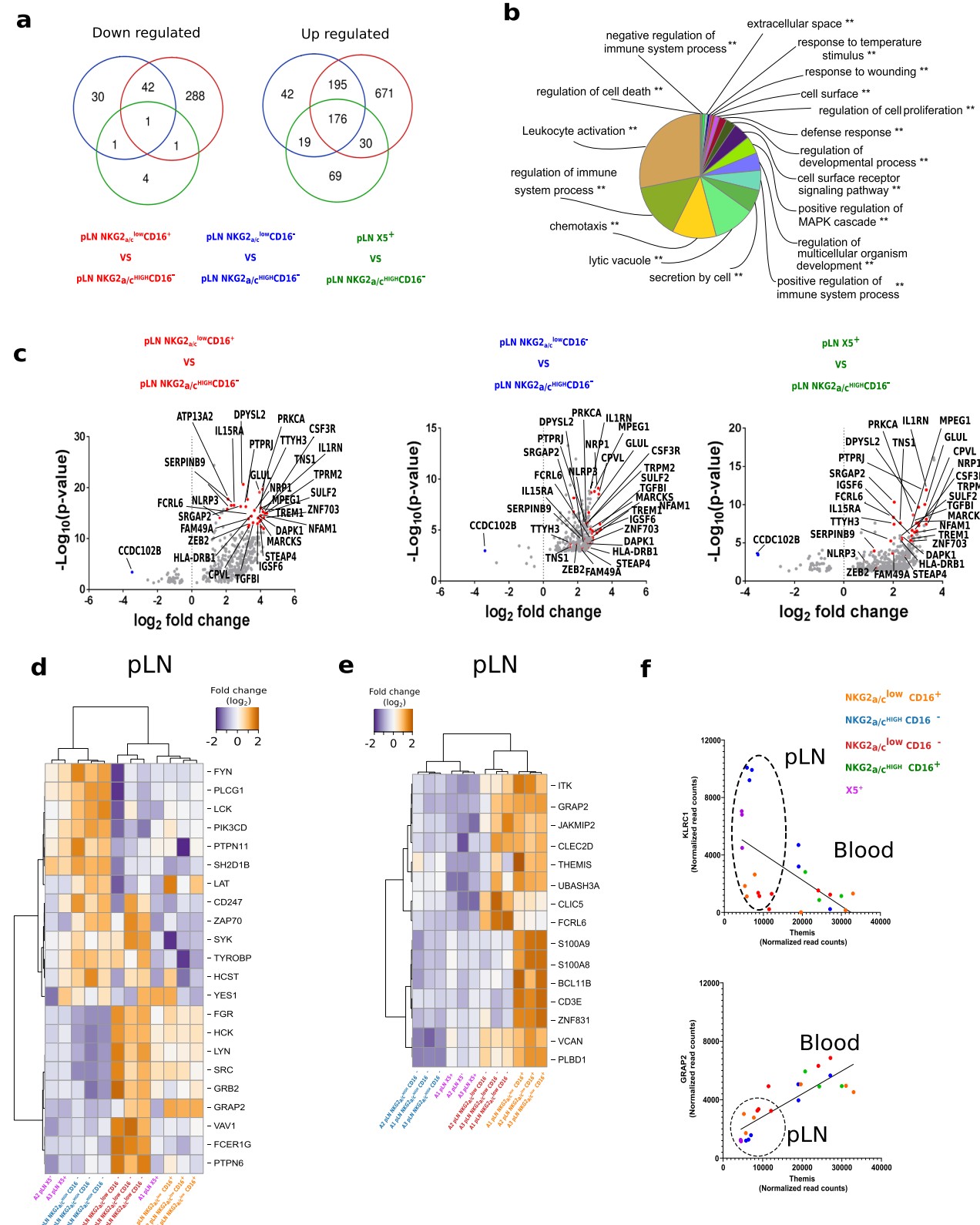

Strong cytotoxic potential and inflammasome activation are compatible with the generation of adaptive NK cells[32–34]. We next analyzed the expression of markers which are so far described as the most specific to adaptive NK cells, meaning the modulation of transcription factors as well as the variable loss of some intracellular adapter signaling molecules[10,35–38] (Fig. 5d).

The NKG2$_{a/c}$high CD16− NK cells did not resemble adaptive NK cells, since they expressed high levels of *LCK*, *CD247* (CD3z), *ZAP70*, *SYK*, *Fyn*, *SH2D1B*, *TYROBP* (DAP12), and *HCST* (DAP10). In contrast, the NKG2$_{a/c}$low CD16+ NK cells in LN fully displayed gene expression profiles characteristic of adaptive NK cells, such as low expression of *LCK, ZAP70, FcεRγ, and SYK*

**Fig. 5 Transcriptomic signatures of adaption and degranulation activity in NK cells from LN during SIVagm infection. a** Venn diagrams generated by the intersection of the list of up- and down-regulated genes with an adjusted $p$-value < 0.05 based on the genome-wide transcriptome. **b** The pie chart shows the enriched Gene Ontology (GO) terms for the 176 genes that were upregulated in $NKG2_{a/c}^{low}$ and $CXCR5^+$ NK cell subsets as compared to $NKG2_{a/c}^{high}CD16^-$ NK cells. The mRNAs were input for ClueGO plugged into Cytoscape for GO enrichment analysis. The 'cellular component', 'biological process', 'molecular function', and 'immune system process' GO terms were selected for this analysis. The two-sided hypergeometric test was used in the statistical inference. The term value corrected with the Bonferroni step down method was applied for $p$-value correlation. The adjusted $p$-value threshold was set to 0.001. The list of the associated genes found for each GO term is given in Supplementary Table S4. **c** Volcano plots that represent up- and down-regulated transcripts of each of the three NK cell subsets when compared to $NKG2_{a/c}^{high}CD16^-$ NK cells. The size of each data point is calculated as $-log10$ ($p$-value) × log2 (FC), with a $p$-value cutoff <0.05. The red dots represent the 30 highest upregulated transcripts in $NKG2_{a/c}^{low}CD16^+$ versus $NKG2_{a/c}^{high}CD16^-$ NK cells. These transcripts were then highlighted on the volcano plots obtained for the two other comparisons. The blue dots represent the transcript coding for CCDC102B, which was downregulated in all three comparisons. On the graph the name of some proteins corresponding to a given transcript are provided. The list of all transcripts upregulated for each comparison is given in the Data Source file. **d, e** Genome-wide transcriptomes were used to define heatmaps of **d** markers linked to ITAM/ITIM signaling pathways and **e** genes strongly upregulated in $NKG2_{a/c}^{low}CD16^+$ NK cells. The orange and blue colors indicate higher and lower levels of transcripts measured in LN NK cell subsets from three chronically infected monkeys. A $p$-value adjustment was performed to account for multiple comparisons and control the false positive rate to a chosen level. $P$ values $\leq 0.05$ were considered statistically significant. In panel **d**, **e** the gene and the name of the proteins corresponding to the transcripts are indicated. Source data been deposited in the Gene Expression Omnibus database; the accession number is GSE140600. **f** Spearman's correlation between *THEMIS* and *KLRC1* (NKG2A; $r = -0.58$, $p = 0.0025$, $n = 24$) or *GRAP2* ($r = 0.84$, $p < 0.000$, $n = 24$) gene expressions. Each dot represents a NK cell subset from an individual animal. Dot circle represents the tissue from which NK cells were isolated.

and high expression of *GRAP2* (Fig. 5d). The $NKG2_{a/c}^{low}CD16^-$ and $CXCR5^+$ NK cells also shared several adaptive NK cell-like features, including low expression of *FcɛRγ* and *VAV1* (Fig. 5d). The levels of *KLRC2* (NKG2C) were negligible when compared to *KLRC1* (NKG2A) transcripts in all NK cell subsets. Altogether, these findings revealed that during SIVagm infection, LN $NKG2_{a/c}^{low}CD16^+$ NK cells, and to a certain extent also $NKG2_{a/c}^{low}CD16^-$ and $CXCR5 +$ NK cells, displayed a transcriptional profile of adaptive NK cells.

We then searched for genes that could be linked to the generation of adaptive NK cells and identified 15 genes that were strongly expressed in $NKG2_{a/c}^{low}CD16^+$ NK cells (Fig. 5e). Among the highly expressed genes were those coding for ZNF831, JAKMIP2, PLBD1, UBASH3, S100A9, LAT (linker for activation of T cell), GRAP2 (GRB2-related adapter protein 2) and THEMIS (Thymocyte-Expressed Molecule Involved in Selection; Fig. 5d, e). *THEMIS* showed a positive correlation with *GRAP2* ($p < 0.004$, $r = 0.74$) and a negative correlation with *NKG2A* ($p < 0.002$, $r = -0.83$) expression (Fig. 5f). Altogether, the data revealed the presence of adaptive NK cells during SIVagm infection expressing low levels of NKG2C and high levels of Themis.

**High MHC-E expression on LN memory CD4$^+$ T cells.** SIVagm-infected AGMs were thus characterized by an expansion of NKG2A$^{low}$ cells in SLT. NKG2A is an inhibitory receptor that binds to a non-classical MHC type I, the MHC-E. Shaping of NK cell function via self-reactive inhibitory NK-cell receptors, such as NKG2A, is a well-described process of education[39–41]. Decreased NKG2A expression is thus indicative of NK cell education. The education via NKG2A involves recognition of peptides presented by MHC-E. MHC-E generally binds self-peptides encoded in the leader sequence (LS) of classical MHC class I molecules, such as the VL9 nonamer peptide[42]. MHC-E can also bind nonamer peptides derived from the stress protein HSP60 as well as pathogen-derived peptides[43–46]. MHC-E is generally expressed at lower levels than other MHC-I molecules and binding of peptides stabilizes MHC-E expression at the cellular surface[42]. The expansion of NKG2A$_{low}$ NK cells in the LN of AGM might be indicative that they became educated to be less inhibited by peptides presented by MHC-E. We evaluated the levels of MHC-E expression on the viral target cells in LN. We found that before SIV infection, among all CD4$^+$ T cells in LN, MHC-E$^+$ cells were found most often among memory CD4$^+$ T cells and T$_{FH}$ cells,

both in MAC and AGM (Fig. 6a, c and Supplementary Fig. 7a–c, f, h). After SIVmac infection in MAC, both the percentage of MHC-E$^+$ memory CD4$^+$ T cells as well as the expression intensity of MHC-E on CD4$^+$ T cells increased and were particularly high for T$_{FH}$ cells (Fig. 6b, d and Supplementary Fig. 7a, b, d, g, i). In contrast, in SIVagm infection, MHC-E expression was only transiently increased in acute phase and the percentage of MHC-E$^+$ T$_{FH}$ cells decreased (Fig. 6a–d and Supplementary Fig. 7). Co-staining of MHC-E expression with SIV RNA could be observed (Fig. 6e). AGM B cell follicles showed accumulation of NK cells close to the MHC-E + cells (Fig. 6f and Supplementary Fig. 8). Therefore, MHC-E expression was most frequent among the preferential target cells for SIV and HIV infection, i.e. memory CD4 + T cells and in particular on T$_{FH}$ cells.

**MHC-E restricted SIV-specific NK cell activity increased after SIV infection in the natural host.** We set up a functional assay to analyze if NK cells show a different ability of MHC-E restricted viral suppressive activity during pathogenic and non-pathogenic SIV infection. To this end, we searched for SIV-derived peptides that would have a high probability for binding to MHC-E and examined whether the SIV genome sequence encodes amino acid sequences similar to the canonical MHC-E binding motif. As described in more detail in the Methods section, we identified in the LS of ENV, nonamer peptides containing the canonical sequence motif of MHC-E binding peptides. These sequences were in addition close to the signal-peptidase cleavage site, therefore showing a potential for the identified peptides for being cleaved in vivo. The identified SIV ENV peptides stabilized HLA-E expression on MHC-I devoid K562 cells stably transduced with HLA-E*0101 (K562-E*0101) (Fig. 7a) indicating their ability for binding to HLA-E. To further analyze the efficacy of binding, competition-binding assays were performed (Fig. 7b). The SIV ENV peptides displaced peptides known to strongly bind to HLA-E (VL9 or HSP60) already at low concentrations (1 µM). The concentrations needed by SIV ENV peptides for displacement of VL9 and HSP60 were in a similar range as those needed by other viral peptides known to efficiently bind to HLA-E (i.e. HIV-1 GAG and HCV peptides[47]) and even lower than for the previously reported SIVmac GAG peptide[48] (Supplementary Fig. 9a). These analyses demonstrate that peptides derived from SIV ENV efficiently bind to HLA-E.

We then analyzed the MHC-E-dependent activity of NK cells toward target cells presenting the SIV ENV peptides via MHC-E.

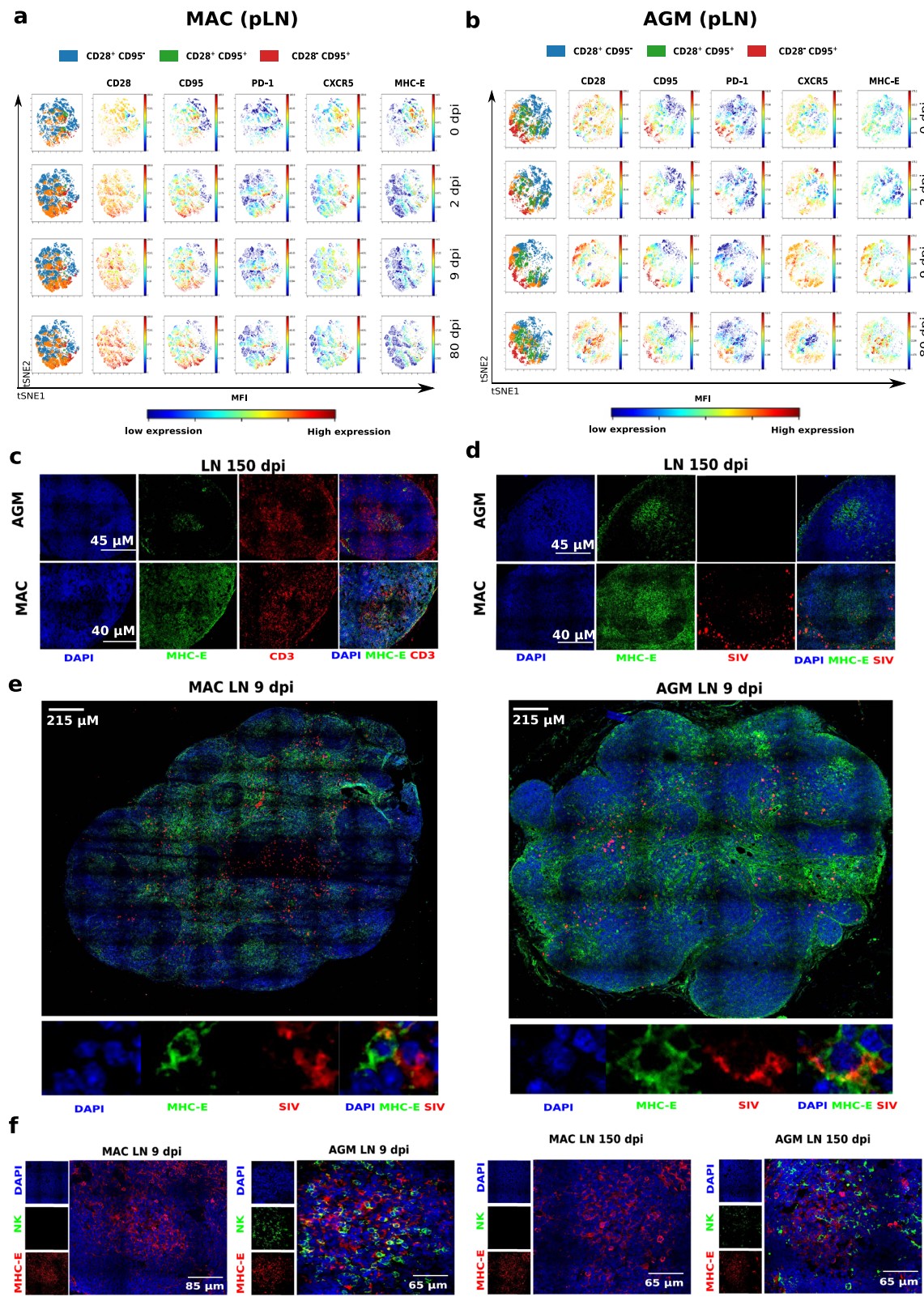

We first studied NK cells from uninfected AGM and MAC. NK cells were co-cultured with K562-E*0101 cells pre-loaded or not with the peptides and their activity was measured by CD107a expression (Fig. 7c, d). As expected, the HLA-I-derived VL9 peptide inhibited NK cell activity, while the HSP60 peptide did not, validating our assay. When pulsing the cells with the SIVmac ENV peptide, NK cell activity from both MAC and AGM were inhibited. The SIVagm peptide displayed an intermediate inhibitory activity on NK cells. We confirmed these results with NK cells from another MAC species (i.e. Indian rhesus macaques; Fig. 7d). The peptides tested did not cover the potential variability of the virus during the infection. However, it should be noted that they were localized in a region which is among the most conserved in the SIV genome[49–51]. Altogether, the data show that

**Fig. 6 Staining of NK, MHC-E + and SIV-RNA positive cells in LN during acute and chronic SIVmac and SIVagm infections. a, b** Illustration of **a** the increase during SIVmac infection and **b** decrease during SIVagm infection of MHC-E expression on memory CD4$^+$ T cells. Results are shown for a representative MAC and a representative AGM among six animals analyzed for each species. The time points analyzed are indicated on the right side of the panels. The viSNE plot was generated according to CD28, CD95, PD-1, CXCR5, and MHC-E expression gated on CD45$^+$CD3$^+$CD14$^-$CD4$^+$ cells in LN using proportional sampling with 54,422 events. The first column represents the distribution of CD4$^+$ T cell subpopulations based on CD28 and CD95 expression: naïve (CD28$^+$CD95$^-$, blue), memory (CD28$^+$CD95$^+$, orange), effector (CD28$^-$CD95$^+$, red), and remaining CD28$^-$CD95$^-$ CD4$^+$ T cells (green). The red and blue colors indicate higher and lower mean fluorescence intensity (MFI) of markers measured in blood NK cells at a given time point. **c–e** Confocal images of LN from MAC and AGM collected in acute or chronic infection. Representative examples are shown for one animal among four studied. Pseudocolors were attributed to each marker (blue for DAPI, green for MHC-E, red for CD3 or SIV RNA, as indicated on the figure). Scale bars are shown on the figure. **c** B cell follicles co-stained for MHC-E and CD3. **d** B cell follicles co-stained for MHC-E and SIV. **e** Whole LN at the top in chronic SIV infection and magnification at the bottom showing co-staining for MHC-E and SIV RNA. **f** Lymph nodes from six AGM and five MAC infected with SIVagm and SIVmac, respectively, were analyzed at days 9 and 150 p.i. One LN per animal was analyzed for each time point. Representative confocal images of LN sections are shown with MHC-E (red), NKG2a (green), and total nucleus (blue).

the NK cells from uninfected animals showed a MHC-E restricted activity that varied dependent on the SIV ENV sequence and that they were more inhibited by the SIVmac ENV than the SIVagm ENV peptides tested.

As a proof of concept of the capacity of the ENV peptides to be processed in primary cells and to be recognized differentially by NK cells depending on their sequence when presented by simian MHC-E, we applied a functional assay using AGM primary CD4$^+$ T cells as targets and full-length infectious SIV as a source for the peptides. Two infectious molecular mutant viruses deriving from the wild-type SIVagm.$_{sab92018}$ backbone were generated. In this backbone, the ENV sequence coding for the nonamer peptide of the wild-type virus (IGIVVIVKL) was replaced either by the sequence coding for the SIVmac$_{239/251}$ peptide (NQLLIAILL) or by the VL9 sequence. The mutant viruses replicated efficiently in primary CD4$^+$ T cells from non-infected AGM (Fig. 7e). The in vitro infected primary CD4$^+$ T cells were co-cultured with autologous NK cells. NK cells strongly suppressed replication of the wild-type SIVagm (Fig. 7e). They controlled the replication of the VL9 mutant less than the wild-type SIVagm, in line with a stronger inhibitory effect of the VL9 peptide. CD4$^+$ T cells infected with the virus containing the sequence coding for the SIVmac ENV peptide escaped NK cell suppressive capacity. These results confirmed a distinct MHC-E restricted NK cell suppressive activity depending on the ENV sequence.

We then reasoned that if NK cells are educated through the NKG2a/MHC-E axis, then the MHC-E restricted viral suppressive capacity of the NK cells after SIVagm infection should be even higher than before SIV infection. We therefore compared the MHC-E dependent suppressive activity of NK cells before and after SIV infection. We exposed NK cells from blood and SLT to K562-E*0101 cells loaded or not with peptides. NK cells from LN of SIVmac-infected MAC showed a drop in their capacity to lyse target cells presenting the SIVmac ENV peptide via MHC-E when compared to NK cells from healthy animals (Fig. 8a). In blood, this capacity was also reduced in both uninfected and SIVmac-infected macaques (Supplementary Fig. 9b). SIVmac controllers did not show stronger ENV-specific NK cell activities than viremic MAC, except for one animal (Supplementary Fig. 9b). In contrast to MAC, NK cells from AGM showed an increased activity after SIVagm infection compared to non-infected AGM in blood and LN (Fig. 8b for LN and Supplementary Fig. 9b for blood).

We tested if the differential changes in NK cell activity after infection were unique for the two SIV peptides or applied to other ENV peptides as well. When using peptides from another SIVagm (SIVagm.sab1) and one more distant SIV (SIVgsn), we observed again strong MHC-E restricted activity with NK cells from LN of SIVagm-infected AGM, while NK cells from SIVmac-infected MAC displayed a weak MHC-E-dependent NK cell

activity in presence of all SIV ENV peptides tested (Fig. 8c, d). We also analyzed NK cells from spleen. As observed for LN, MHC-E restricted activity of NK cells from spleen showed trends for decreased activity toward cells presenting ENV peptides from SIVmac$_{239/251}$, SIVagm.$_{sab}$, SIVagm.$_{tan}$, and SIVgsn in MAC after SIVmac infection, while this activity increased after SIVagm infection in the presence of SIVagm.sab$_{92018}$, SIVagm.$_{sab1}$, and SIVmac$_{239/251}$ peptides (Supplementary Fig. 9c). Altogether, our results revealed that the MHC-E restricted suppressive NK cell activity decreased in SIV-infected MAC in SLT, while it was enhanced in SIV-infected AGM.

**Discussion**

Previous studies mostly focused on circulating NK cells during HIV/SIV infection, and only a few studies on tissue NK cells have been reported[13,14,20,52,53]. Here, we designed a study comparing NK cells associated with a strong viral control in SLT (SIVagm infection) to NK cells lacking the capacity of an efficient viral control (SIVmac infection). Our results show that SIV infection in the natural host induced terminally differentiated NKG2A$^{low}$CD16$^+$NK cells in LN displaying a strong degranulation activity and a transcriptional profile of adaptive NK cells. Consistently, the MHC-E restricted NK cell activity against target cells presenting SIV peptides was increased after SIVagm infection. In contrast, the process of terminal NK cell differentiation was blocked in LN during SIVmac infection of MAC in early infection, despite the generation of a highly diverse NK cell pool. Our results uncover that NK cells are educated to efficiently kill SIV-infected cells in SLT during non-pathogenic SIVagm infection while this education is not occurring in pathogenic SIV infection.

The notion that NK cell differentiation is the result of a continuous process that can adapt in response to a viral infection is relatively recent[6,7,30,54–56]. By analyzing distinct NK cell subsets from AGM and MAC at the transcriptomic and protein level we demonstrate here that NKG2A combined with CD16 defines distinct stages of NK cell differentiation in monkeys. This provides a tool for preclinical studies in NHP for the investigation of other infections and for vaccine studies in general.

The NKG2$_{a/c}$$^{low}$CD16$^+$ NK cells clearly displayed the most differentiated profile and strong cytotoxic activity. We also observed cytotoxic activity for the NKG2$_{a/c}$$^{low}$CD16$^-$ and CXCR5$^+$ NK cells. CD16 can be quickly shed from the NK cell surface upon activation[57]. Under these conditions, NKG2$_{a/c}$$^{low}$CD16$^-$ and CXCR5$^+$ NK cells might be a mixture of less differentiated NK cells together with terminal differentiated NKG2$_{a/c}$$^{low}$CD16$^+$ cells that exert the viral control. The NKG2$_{a/c}$$^{low}$CD16$^+$CXCR5$^+$ NK cells might be responsible for the viral control in B cell follicles and the NKG2$_{a/c}$$^{low}$CD16$^+$CXCR5$^-$ NK cells in the T cell zone of lymphoid tissues. Future studies will need to address which of the activatory NK cell receptors are involved in the suppressive activity.

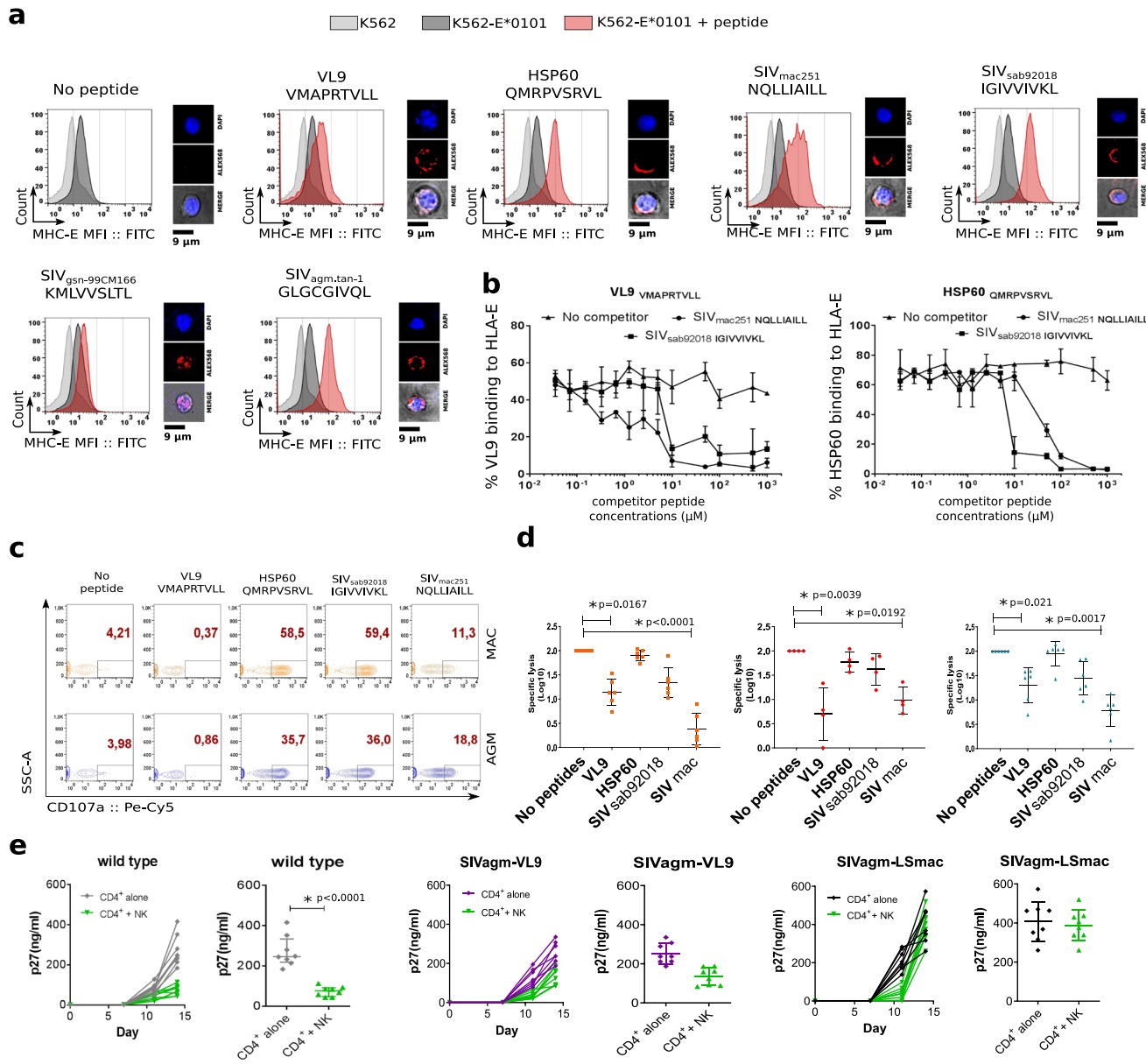

**Fig. 7 Analysis of MHC-E restricted NK cell activity. a** Each histogram shows HLA-E expression after loading K562-E*0101 cells with the different peptides. Light gray curves indicate the mean fluorescent intensity (MFI) of HLA-E on K562 cells (control), the dark gray curves the MFI of HLA-E on K562-E*0101 cells without peptides, and the red curves the MFI of HLA-E on K562-E*0101 cells loaded with the given peptide. VL9 and HSP60 are control peptides well known to bind to MHC-E and deriving respectively from HLA-I and heat-shock protein. The SIVmac251 and SIVmac239 peptide sequences were identical. One representative experiment out of more than three is shown. On the right side are shown the localization of the peptide stained with Alexa 568 (red). Nuclei were stained with DAPI (blue). **b** Competition assay for binding to HLA-E. K562-E*0101 cells were loaded with biotin-A2 VL9 (left) or HSP60 peptides (right). An increasing concentration of the indicated unlabeled competitor peptides were introduced into the culture. HLA-E-bound biotinylated peptide was measured at the cell surface. line represent the median of three independent experiments. Error bars represents the range. **c** Analysis of HLA-E dependent activity of NK in presence of K562-E*0101 cells loaded or not with peptides. Activity was evaluated by CD107a cell surface expression. Dot plots show representative examples for one animal per species. **d** HLA-E dependent activity of NK cells from blood of six uninfected cynomolgus MAC (orange squares), four uninfected rhesus MAC (red circles) and six uninfected AGM (blue triangles) in the absence or presence of peptides, as indicated. The black line indicates the median and error bars the interquartile range. Each symbols represents a unique monkey. For Groups comparisons two-sided Wilcoxon signed-rank test with Bonferroni correction were used ($n = 5$). P values of less or equal to 0.05 were considered statistically significant. Asterix indicate significant change when compared to the base line. Exact P values are provide on the graphs. **e** SIV suppressive capacity of autologous NK cells depending on the ENV nonamer sequence in the infectious virus. The panels show in vitro replication levels of SIVagm.sab92018 (gray), SIVagm.sab92018 mutant with the sequence coding for the VL9 peptide (purple) and the SIVagm.sab92018 mutant with the sequence coding for the SIVmac nonamer peptide (black) in primary CD4+ T cells from 8 uninfected AGM cultured alone or in presence of autologous NK cells (green). For each of the three viruses, left panels show longitudinal viral replication pattern (days 0, 7, 11 and 15 p.i.) and right panels viral replication at the end time-point of the culture (day 15 p.i.). The y-axis reports viral replication level as quantified by SIV p27. The black line indicates the median and error bars the interquartile range. Significant difference between the two conditions was determind using using two-tail Mann–Whitney per experiment. Asterix indicate a statistically significant difference. The exact P value is provided.

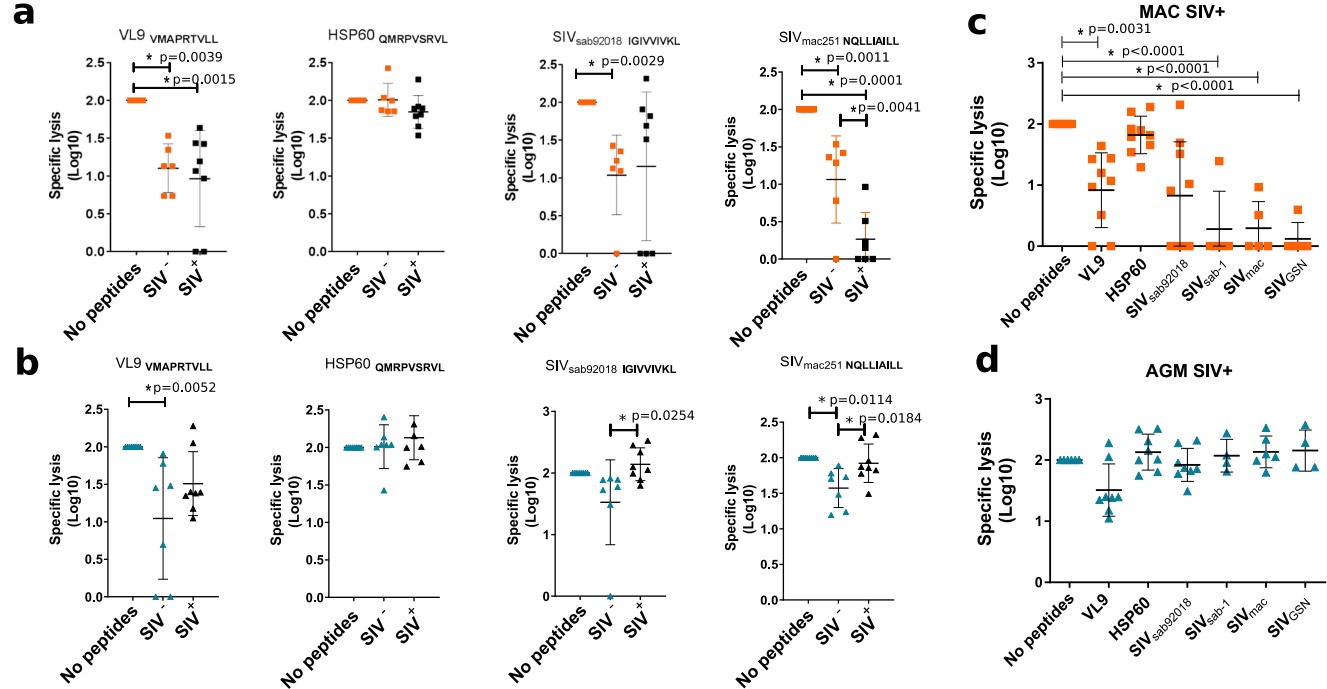

**Fig. 8 Differences in MHC-E restricted activity of secondary lymphoid tissue NK cells between natural and heterologous hosts. a, b** HLA-E dependent cytotoxic activity of NK cells isolated before infection and in chronic infection from LN of MAC (orange), and AGM (blue). K562-E*0101 were loaded or not with peptides. Uninfected MAC are indicated in orange squares and uninfected AGM in blue triangles. SIV-infected animals are indicated in black. The black line indicates the median and error bars the interquartile range. Each symbols represents a unique monkey. For Groups comparisons two-sided Wilcoxon signed-rank test with Bonferroni correction were used ($n = 3$). P values of less or equal to 0.05 were considered statistically significant. Asterix indicate significant change when compared to the base line. Exact P value are provided on the graphs. **c, d** HLA-E dependent cytotoxic activity of NK cells isolated from spleen of chronically infected **c** MAC and **d** AGM against distinct SIV ENV peptides. The black line indicates the median and error bars the interquartile range. Each symbols represents a unique monkey. For Groups comparisons two-sided Wilcoxon signed-rank test with Bonferroni correction were used ($n = 7$). P values of less or equal to 0.05 were considered statistically significant. Asterix indicate significant change when compared to the base line. Exact P values are provided on the graphs.

Also, while our study demonstrates a superior MHC-E restricted NK cell activity in the natural host, this might not be the unique mechanism of viral suppression, since terminally differentiated cells also up-regulate for example ADCC-mediated killing capacity[58].

The reasons why $NKG2_{a/c}^{low}$ NK cells expanded in SIVagm infection in AGM but not in SIVmac infection in MAC are unclear. These cells did not show increased proliferation and it is unlikely that they result from increased trafficking to lymphoid tissues, since we reported that LN homing receptors decrease on NK cells from AGM in SIVagm infection similarly to SIVmac and HIV infection[20]. It is therefore more likely that this expansion of $NKG2_{a/c}^{low}$ NK cells is due to NK cell differentiation in situ[15]. A major difference between the two infections consists in the persistence of an inflammatory environment in SLT during SIVmac infection while inflammation is resolved in SIVagm infection[19,53]. Systemic inflammation has been shown to drive the expansion of IFN-γ producing NK cells in mice[59,60]. Similarly, it has been recently shown that inflammatory cytokines drive the expansion of CD94−CD56bright NK cells with an increased capacity to produce IFN-γ in HIV infection that could themselves contribute to the inflammation[61]. These cells might correspond to the $NKG2_{a/c}^{high}CD16^{−}$ NK cells that were the predominant NK cell subset in LN of SIVmac-infected MAC and expressing IFN-γ. Notably, the terminally differentiated NK cells in AGM LN expressed high levels of IL15R and IL21R. Higher levels of IL-15 and IL-21 in the local environment during SIVagm infection as compared to SIVmac infection might allow NK cell terminal differentiation. Considering that AGMs are ancient hosts of SIV,

it is possible that the long-term co-existence in AGM selected a specific tissue environment in response to SIV infection that allows proper education of NK cells. By driving NK cells toward a terminal differentiated, cytotoxic phenotype, this would allow the presence of NK cells with capacity to kill infected cells in LN, while avoiding the inflammation associated with NK-cell mediated IFN-γ production.

The distinct MHC-E expression profiles on CD4 + T cells in LN between SIVmac and SIVagm infection might be determined by host and/or viral factors[59–63]. The frequency of productively infected cells in chronic SIVagm infection in SLT is too low to explain the decreased levels of MHC-E solely by virus-mediated down-regulation of MHC-E or NK-cell mediated killing of MHC-E + cells. A distinct inflammatory environment between SIVmac and SIVagm infections, in particular higher levels of IFN-γ in SIVmac than SIVagm infection[62–66], could be one reason for the higher MHC-E expressions in SIVmac infection.

It has been shown that genetic factors determining MHC-E expression levels strongly affect viral load in HIV-infected individuals[67]. This is in line with reports on the importance of MHC-E in SIV control. It has previously been reported that MHC-E-dependent NK cell suppressive capacity is dependent of the nature of the peptide loaded by MHC-E[46,47,68,69] and that peptides derived from SIVmac can inhibit NK cells[46,47,70]. Only few MHC-E binding viral peptides have been described so far[69,71]. We define here a lentiviral region (ENV-LS), that can code for peptides with the capacity to bind to MHC-E. This region might thus serve as a critical target for NK cell directed

HIV vaccines, cure strategies, or other therapeutic modalities. Further analyses are needed to identify the sequences of peptides bound to MHC-E in vivo. Of note, the target cells and peptides used in our assays were the same before and after SIV infection, clearly demonstrating that the increased MHC-E restricted activity of NK cells in SIVagm infection must be due to changes at the level of the NK cells themselves.

Examples are known where NKG2A-dependent NK cell education is a determining factor for NK cell responses in infectious conditions, such as CMV infection in humans[5,40,58,72]. The increased functional activity here was not related to stronger expression of the activating NKG2C receptor. The NK cells that expanded thus are not identical to the NK cells that are induced upon CMV infection[73–75]. The signaling pathways of adaptive NK cells are poorly understood. The adaptive NK cells here expressed high level of transcripts coding for THEMIS, LAT, and GRAP2. THEMIS is known to play a role in T cells, where it neutralizes major negative regulators (SHP-1, GRB2) of proximal TCR signaling[76–78]. NKG2A induces the activation of SHP1 or SHP2, which blocks the function of LAT[37]. By increasing THEMIS, one could speculate that SHP1 and 2 are blocked, releasing LAT despite the presence of NKG2A. However, whether THEMIS is functional in NK cells is unclear and will need to be addressed in the future.

In conclusion, we demonstrate that SIVagm infection induces the expansion of terminally differentiated NKG2a$^{low}$ NK cells in SLT displaying an adaptive transcriptional profile and increased MHC-E-restricted cytotoxicity in response to SIV ENV peptides while expressing little IFN-γ. Such NK cell differentiation and adaptation was lacking in SIVmac-infected macaques. This study describes and provides mechanistic insights into functional differentiation of NK cells during a controlled viral infection in a tissue of crucial importance, the LNs, with key implications for an optimized guidance on NK-cell-based immunotherapies toward HIV cure.

## Methods

**Monkeys**. Twelve African green monkeys (Caribbean *Chlorocebus sabaeus*, AGM), 9 cynomolgus macaques (*Macaca fascicularis*, cynMAC) and 18 Indian rhesus macaques (*Macaca mulatta*, rhMAC) were studied. The AGM and cynMAC were housed at the IDMIT Center (Fontenay-aux-Roses, France) and the rhMAC at the German Primate Center (DPZ). All experimental procedures were conducted in strict accordance with the international European guidelines 2010/63/UE on the protection of animals used for experimentation and other scientific purposes and with the German and French law (French decree 2013-118), and the German Animal Welfare Act and also with the recommendations of the Weatherall report. The IDMIT center complies with the Standards for Human Care and Use of the Office for Laboratory Animal Welfare (OLAW, USA) under OLAW Assurance number A5826-86. Monitoring of the monkeys was under the supervision of the veterinarians in charge of the animal facilities. Animal experimental protocols were approved by the Ethical Committee of Animal Experimentation (CETEA-DSV, IDF, France; Notification 12-098 and A17-044). The pVISCONTI study was approved and accredited under statement number A15-035 by the ethics committee "*Comité d'Ethique en Expérimentation Animale du CEA*", registered and authorized under Number 2453-2015102713323361v2 by the French Ministry of Education and Research. The study at DPZ was approved by the Lower Saxony State Office for Consumer Protection and Food Safety and performed with the project licences 33.19-42502-04-12/0820 and 33.19-42502-04-17/2500. The DPZ has the permission to breed and house non-human primates under license number 392001/7 granted by the local veterinary office and conforming with § 11 of the German Animal Welfare act.

The animals were healthy and seronegative for SIV, type D retrovirus, and simian T-cell lymphotropic virus type 1 at the time of infection and were housed in single cages within level 3 biosafety facilities after infection. At the inclusion in the study the average weight of the monkeys was between 3 and 6 kg. All monkeys were young adults with an average age of 3–5 years at inclusion. Both males and females were used (60% females and 40% males for each species). Because H6 haplotypes are notably associated with viral control in cynomolgus macaques, macaques with H6 haplotype were excluded from this study. The sample size varied between 3 and 9 monkeys per group ($n = 6$ in most experiments), chosen according to the tripartite harmonized International Council for Harmonization of Technical Requirements for Pharmaceuticals for Human Use (ICH) Guideline on

Methodology (previously coded Q2B). Sample collection was performed in random order. The investigators were not blinded while the animal handlers were blinded to group allocation.

**SIV infections**. Briefly, Monkeys were sedated with Ketamine Chlorhydrate (Rhone-Mérieux, Lyons, France) before handling. African green monkeys (Caribbean *Chlorocebus sabaeus*, AGM) and cynomolgus macaques (*Macaca fascicularis*, cynMAC) were infected intravenously with 250 median tissue culture infectious dose (TCID$_{50}$) of SIVagm.sab$_{92018}$ and 5000 or 1000 median animal infectious dose (AID$_{50}$) of SIVmac$_{251}$ respectively. The rhesus macaques (*Macaca mulatta*, rhMAC) were infected intravenously with 300 TCID50 SIV$_{mac239}$. The viremia levels are shown in Supplementary Fig 2. All cynMAC where viremic. Five rhMAC were used as uninfected controls. Six out of the 13 SIVmac-infected rhMAC were viremic and seven were SIV Controllers (SIC)[79]. If not specified otherwise in the text, the species of the macaques used was cynomolgus.

**Tissue collections and processing**. Whole venous blood was collected in ethylenediaminetetraacetic acid (EDTA) tubes. Peripheral blood mononuclear cells (PBMCs) were isolated by Ficoll density-gradient centrifugation. Biopsies of peripheral lymph nodes (pLN) were performed by excision. Other tissues were collected at autopsy. After careful removal of adhering connective and fat tissues, LN and spleen cells were dissociated using the gentlemacS™ Dissociator technology (Miltenyi Biotec, Germany). The cell suspension was subsequently filtered through 100- and 40-μm cell strainers, and cells were washed with cold phosphate-buffered saline (PBS). Cells were either immediately stained for flow cytometry or cryopreserved in 90% foetal bovine serum (FBS) and 10% dimethyl sulfphoxide (DMSO) and stored in liquid nitrogen.

**Plasma viral load**. The viral RNA copy numbers in plasma and the cell-asssociated viral DNA and RNA of the animals were quantified by real-time PCR assay[80,81,82]. The primers used in the assays are given in Supplementary Table 2. Viral RNA was measured by qPCR in duplicate. SIVagm and SIVmac251 products of T7 transcription from plasmids were used as standards to calculate SIV RNA copy numbers. 18S ribosomal RNA and CCR5 DNA quantification were used for normalization. Sample preparation, enzyme mix preparation and PCR setup were performed in three separate rooms to avoid PCR contamination. Negative controls were used to exclude sample contamination. The reactions were run in an ABI Prism 7500 with one cycle at 95 °C (15 min) followed by 40 cycles at 95 °C (15 s) and 55 °C (1 min). The cut-off value was 10 viral copies or below per ml of plasma.

Viral RNA from rhesus macaques was extracted from 140 μl plasma using QiaAmp viral RNA isolation mini kit (Qiagen). For qRT-PCR, 8.5 μl RNA-solution were reverse transcribed and amplified using TaKaRa Prime-Script-One-Step-RT-PCR kit (TaKaRa Bio Europe). Reverse transcription was performed at 45 °C for 5 min, amplification was started by an initial denaturation step at 95 °C for 10 s, followed by 45 cycles of denaturation for 5 s at 95 °C and annealing and elongation at 60 °C for 30 s using the Rotor-Gene Q apparatus and software (Qiagen). The cut-off value was 40 viral copies per ml plasma.

**Construction of the infectious molecular clones of SIVagm**. Two infectious recombinant molecular clones were derived from the SIVagm.sab92018 clone (SIVagmSab92018ivTF) which itself derived from a SIVagm virus that was isolated from a naturally infected AGM (92018) and has never been in vitro cultured before cloning[83,84]. The recombinant clones harbor the ENV LS sequence coding either for the nonamer peptide of SIVmac$_{239/251}$ (NQLLIAILL; aatcagctgcttatcgc-catcttgctt) or VL9 (VMAPRTLL; gtgatggcgccgcgcaccctgctgctg). The mutants were generated by replacement of the wild-type ENV LS sequence of SIVagm.sab using site-directed mutagenesis (cat:E0554; NEB New England Biolabs). The primers used to construct the different mutants are list in Supplementary Table 2.

**Polychromatic flow cytometry**. Flow cytometric analysis was performed on fresh or Frozen, PBMCs and pLN cells. Samples were stained utilizing standard procedures employing clones of anti-human monoclonal antibodies (mAbs) that we have shown to be cross-reactive in the NHP species used in the study. The antibody clones and references are listed in Supplementary Table 3. The anti-NKG2A antibody used recognizes both NKG2A and NKG2C on simian cells[80]. Gating strategy used for NK and CD4 T cells analysis are shown in Supplementary Fig. 1. Flow cytometry acquisitions were done on a LSRFortessa (BD Biosciences). Intracellular staining was performed using BD Cytofix/Cytoperm™. The data were further analyzed using FlowJo 10.4.2 software (FlowJo, LLC, Ashland, OR, USA). Multiparametric analyses were performed using SPICE (version 5.1). *t*-SNE was performed with the *cytobank* (Cytobank, Inc.), using 2000 iterations and a perplexity of 60.

**Immunofluorescence staining**. K562-E*0101 cells were incubated 16 h with biotinylated peptides. Cells were then fixed with PFA 4% and staining was performed using DAPI and Streptavidin coupled with Alexa 568 for 1 h at 37 °C followed by three washing steps. Cells were analyzed using a Nikon microscope.

Appropriate filters for immunofluorescence analysis of labeled cells were used and images were imported into ImageJ software.

Fresh pLN tissues were embedded and snap frozen in optimum cold temperature compound (OCT) and 10 µm frozen sections were stained using unconjugated primary antibodies followed by appropriate secondary antibodies conjugated to Alexa 488 (green), Alexa 568 (red; Molecular Probes, Eugene, OR; dilution: 1/2000). The primary Abs used in this study are anti-NKG2a (Epitomics (T3308), Ag Retrieval:10 mM citrate buffer (pH 6) 98 °C 20 min, dilution: 1/200); anti CD3 (AbD Serotec (MCA 1477), Ag Retrieval:10 mM citrate buffer (pH 6) 98 °C 20 min; dilution: 1/300), and anti HLA-E (MEM-E/06 (ab3984), Ag Retrieval:1 mM EDTA buffer (pH 8) 98 °C 20 min; dilution: 1/200). FISH assay combined with immunofluorescent staining was performed as previously described[20]. Stained slides were incubated with 100–200 µL of ice cold methanol and 5% acetic acid, placed at -20 °C for 10 min and then washed. Image analysis was performed using a Leica TCS SP8 confocal microscope equipped with white lasers (Leica Microsystems, Exton, PA).

**Generation of a MHC-E⁺K562 cell line**. K562 were used as packaging cells to generate a lentivirus coding for HLA-E*0101 under the control of a CMV promotor (pLenti-GIII-CMV-RFP-2A-Puro lentiviral vectors (ABMGood)) according to manufacturer's instructions. Lentiviral particles were concentrated using Polybrene and Chondroitin Sulfate and viral pellets resuspended in one hundredth of the original volume in DMEM medium.

Before transduction, the coated plates were washed with dH₂O twice and then exposed to UV light for 45 min. K562 cells obtained from the American Type Culture Collection (ATCC) were seeded at a density of $2 \times 10^4$ cells per well (in coated and uncoated plates) and incubated at 37 °C in a humidified 5% CO₂ incubator overnight. Cell culture medium in plates was removed leaving 100 µl medium in each well. Thirty minutes before transduction, the lentiviral vector was thawed on ice and 8 µg Polybrene was added to 100 µl of the virus in the tube. Viral vector-Polybrene complexes were added to each well, in coated and uncoated plates, at distinct multiplicities of infection (5, 10, 15, and 20 MOI). Plates were shaken gently and placed back in the incubator and incubated at 37 °C and 5% CO2. Six hours after transduction, K562 cells were centrifuged at 800×g for 10 s and cultured for 70 min at 32 °C. After that, the cells were returned to the plates and incubated overnight. Cells were then seeded into 96 wells plated at the concentration of one cell per well in order to generate unique clones. After 5 weeks, transduction efficiency was evaluated by measuring the percentage of RFP-expressing cells by flow cytometry. One clone was selected, amplified, and used for all the experiments.

**Isolation of CD4 + T cells and NK cells and Cell sorting of NK cell sub-populations**. CD4⁺ T cells and NK cells from fresh or frozen PBMC, pLN and spleen were isolated using,a positive selection with respectively, the anti-human CD4 (CD4 MicroBeads, human Miltenyi Biotec (USA)), or the anti-NKG2a/c monoclonal antibody-PE conjugated (clone Z199 *Beckman Coulter (USA)*) reveals by anti-PE microbeads (Miltenyi Biotec (USA)), the staining and the positive selection of the cells was done according to the supplier's instructions.

Eight-color panels were used to phenotype, surface stain and sort NK cells from blood and pLNs from chronically infected AGMs. Cells were thawed in 20% FBS-containing media supplemented with benzonase nuclease, and counts and viabilities were performed (Life Technologies). Cells were washed and stained with Aqua Live/Dead stain (Molecular Probes). Cells were washed and blocked using normal mouse IgG (Caltag). For the NKG2a/c^HIGH/low^CD16⁺/⁻phenotyping panel, PBMCs and pLNs were surface stained for CD3 (SP34.2, 1:10 dilution; BD), CD8 (BW135/80, 1:20 dilution; Miltenyi), CD16 (3G8, 1:20 dilution; Beckman Coulter, Inc.), NKG2a/c (Z199, 1:20 dilution; Beckman Coulter, Inc.), CD20 (2H7, 1:20 dilution 1/20; Biolegend), and CD14 (M5E2, 1:25 dilution; BD). Post-staining, cells were washed, filtered, and sorted on a FACS ARIA II (BD). Cells were directly collected in a lysis buffer that contained TCEP. The purity of the cells was >97%.

**Cell culture**. K562 (human HLA class I–negative erythroleukemia) (ATCC® CCL-243) as well as HLA-E transduced K562 cells were maintained in RPMI 1640 (Life Technologies) supplemented with 10% heat-inactivated foetal calf serum (FCS), 2 mM, l-glutamine, 100 U/ml penicillin, and 100 µg/ml streptomycin.

CD4⁺ T cells were maintained in RPMI 1640 with Glutamax (Life Technologies) supplemented with 10% heat-inactivated FCS, 2 mM, l-glutamine, 100 U/ml penicillin, 100 µg/ml streptomycin, and 100 IU/ml of IL-2. NK cells were cultured in the same medium supplemented with 10 ng/mL of IL-15.

**Infection of primary CD4⁺ T cells with SIVagm viruses**. CD4⁺ T cells from SIV-negative AGMs rested overnight in the medium described above supplemented with IL-4 (20 µg/ml). They were stimulated with anti-CD4 and -CD3. After 2 days, $5 \times 10^6$ cells CD4⁺ T cells were exposed to virus (4 ng of p27 capsid antigen) for 1 h. Subsequently, cells were washed extensively to remove cell-free virus and maintained in IL-2 medium. Virus production in culture supernatants was monitored at regular intervals by SIV p27 antigen capture assay (Zeptometrix) according to the manufacturer's instructions.

**Design of SIVagm and SIVmac ENV peptides with potential for efficient binding to HLA-E**. We searched for SIV-derived peptides that would have a high probability for binding to MHC-E and examined whether the SIV genome sequence encodes amino acid sequences similar to the canonical MHC-E binding motif found in the leader sequence (LS) of classical MHC class I[42] and some cellular proteins such as the stress protein HSP60[43]. Since retroviral ENV proteins contain a LS, we focused on that region and screened the ENV LS sequences from eight distinct SIV. We identified 28 peptides showing similarities with the MHC-E permissive motif (Supplementary method Fig. 2a). We used a prediction algorithm to examine these 28 Env LS peptides as well as two control peptides (VL9 (VMAPRTVLL) derived from HLA-B*0701 and HSP60 (QMRPVSRVL)) for their binding affinity to HLA-E*0101 and HLA-E*0103, the only two functional HLA-E alleles in humans (Supplementary method Fig. 2b and Supplementary method Table 3)[85,86]. We selected for each SIV the ENV peptide with a high probability for binding to MHC-E (Supplementary method Fig. 2c and Supplementary method Table 3). The peptides selected for SIVmac239 and SVmac251 were identical to each other. The MHC-E peptide-binding groove is conserved among primates and it has been shown that human and MAC MHC-E are promiscuous and able to bind identical peptides[47,87]. Three-dimensional analyses indicated that AGM MHC-E was also conserved with respect to human and other non-human primate MHC-E (Supplementary method Fig. 2d).

**Peptides and HLA-E stabilization assay**. Synthetic peptides biotinylated or not were purchased from *Proimmune* (United Kingdom) and *Proteogenix* (France) and dissolved in DMSO at the concentration of 2 mg/ml. The peptides used were VL9 (VMAPRTVLL), HSP60 (QMRPVSRVL), SIV_MAC239/251_ (NQLLIAILL), SIVagm_SAB92018_ (IGIVVIVKL), SIVagm_SAB-1_ (SGCWSLVWL), SIVagm_TAN1_ (GLGVGIVQL), and SIV_GSN_ (KMLVVSLTL). K562-E*0101 cells were incubated with synthetic peptides (3–300 µM) at 37 °C for 15–20 h in serum-free AIM-V medium (GIBCO BRL) at a concentration of $1–3\ 10^6$ cells/ml. Control cultures were kept at 37 °C for over 16 h without peptides. Cells were then harvested, washed in PBS, and cell-surface expression of HLA-E was determined by incubation with PE-conjugated anti-HLA-E antibody, washed twice with PBS and fixed in 100 µl of Cytofix (BD Biosciences). Cells were acquired using a LSR II (BD Biosciences), and FlowJo software (version 9.6.4, Tree Star, Ashland, OR) was used for all analyses. Results were expressed either directly as mean fluorescent intensity (MFI).

**Peptide competition assay**. Peptide-binding assays were performed according to methods described in the literature (36). In brief, K562-E*0101 cells were incubated overnight (18 h) at 37 °C with biotinylated peptides (20 µM) and dilutions of non-biotinylated competitor peptide were added to the culture binding buffer (0.01% Nonidet P-40, 10 nM Citrate-phosphate buffer and Protease inhibitor cocktail (Roche, Nutley, NJ). Cell surface biotinylated peptides were determined by adding alexa-488 labeled Streptavidin to each sample well. The expression of biotinylated peptides was assessed by flow cytometry. Cells were acquired using a LSR II (BD Biosciences), and FlowJo software (version 9.6.4, Tree Star, Ashland, OR) was used for all analyses. Results were expressed either directly as mean fluorescent intensity (MFI).

**NK cell cytotoxic in vitro assay**. NK cell activity was determined through expression of cell surface CD107a[88,89]. NK cells were cultured overnight in the presence or absence of the MHC-class-I-devoid target cell line K-562 at an effector: target ratio of 1:5 for a total of 6–8 h in the presence of anti-human CD107a. Monensin (BD GolgiStop, BD Biosciences, Franklin Lakes, NJ, USA) was added 1 h after setup of the co-culture, followed by an additional 5 h of incubation at 37 °C in a humidified atmosphere with 5% (v/v) CO₂. Cells were then washed with PBS and stained for viability and expression of CD3, CD14, CD16, CD8, CD20, NKG2a/c, CD16, and CD107a.

**MHC-E-dependent NK cell viral suppressor assays**. K562-E*0101 cells were incubated with 50 µM of a given peptide at 26 °C for 15–20 h. NK cells were co-cultured with $2 \times 10^4$ K562-E*0101 target cells pulsed or not with peptides at a NK cell/target cell ratio of 5:1. Target cells were in parallel cultured in the absence of NK cells. Anti-CD107a antibody was added at the start of the assay, and GolgiStop and GolgiPlug (both BD Biosciences) were added 1 h after start of the stimulation. After 6 h of culture, cell suspensions were stained for viability and cytotoxic activity was analyzed through measurement of CD107a expression by flow cytometry. Percent activity was calculated as described above.

CD4⁺ T cells ($5 \times 10^6$ cells) activated for 2 days as described above were co-cultured with autologous NK cells (ratio 1:1) and from the time of virus exposure on.

**RNA extraction, library preparation, and sequencing**. RNA was isolated from the sorted NK cells using the RNeasy® Mini Kit (205113, Qiagen). RNA integrity was verified with the Agilent Bioanalyser. DNase-treated RNA was treated for library preparation using the Truseq Stranded mRNA Sample Preparation Kit (Illumina, San Diego, CA), according to manufacturer's instructions. An initial poly(A) RNA isolation step (included in the Illumina protocol) is performed on

10 ng of total RNA to keep only the polyadenylated RNA fraction and remove the ribosomal RNA. A step of fragmentation is then performed on the enriched fraction, by divalent ions at high temperature. The fragmented RNA samples were randomly primed for reverse transcription followed by second-strand synthesis to create double-stranded cDNA fragments. No end repair step was necessary. An adenine was added to the 3′-end and specific Illumina adapters were ligated. Ligation products were submitted to PCR amplification. The obtained oriented libraries were controlled by Bioanalyzer DNA1000 Chips (Agilent, # 5067-1504) and quantified by spectrofluorimetry (Quant-iT™ High-Sensitivity DNA Assay Kit, #Q33120, Invitrogen). Sequencing was performed on the Illumina Hiseq2500 platform to generate single-end 100 bp reads bearing strand specificity.

**Homology modeling of MHC-E and model analysis**. Since there is no X-ray structure available for MHC-E from non-human primates, structures for rhesus, cynomolgus, African green monkey, baboon, and chimpanzee proteins were modeled with the software MODELLER using the crystal structure of HLA-E*01:03 (PDB code 3BZF) as template[90]. MHC-E amino acid conservations were analyzed with the ConSurf server[91]. Illustrations were rendered with PyMol.

**Bioinformatic analysis of the genome-wide sequence data**. Bioinformatic analyses were performed using the RNA-seq pipeline from Sequana[83]. Reads were cleaned of adapter sequences, and low-quality sequences were removed using cutadapt version 1.11[92]. Only sequences ≥25 nucleotides (nt) in length were considered for further analysis. STAR version 2.5.0a, with default parameters, was used for alignment on the reference genome (*Chlorocebus sabaeus*, from Ensembl release 90). Genes were counted using featureCounts version 1.4.6-p3[93] from Subreads package (parameters: -t gene, -g ID and -s 1).

Data were analyzed using R version 3.4.3 and the Bioconductor package DESeq2 version 1.18.1[94]. Normalization and dispersion estimation were performed with DESeq2, using the default parameters, and statistical tests for differential expression were performed by applying the independent filtering algorithm. A generalized linear model, including the monkey identifier as a blocking factor, was used to test for the differential expression between the biological conditions. For each pairwise comparison, raw p values were adjusted for multiple testing according to the Benjamini and Hochberg (BH) procedure[95]. Genes with an adjusted $p$ value < 0.05 were considered differentially expressed.

Analyses and vizualization of pathways associated with differentially expressed genes were performed using ClueGO. Both groups of genes (up- and downregulated, $p$ value < 0.05) were used as dual input for GO and pathway annotation networks of the expressed genes and proteins pathway enrichment analysis[96]. Each list was used to query the Kyoto Encyclopedia of Genes and Genomes (KEGG), GO-biological function database and Wiki pathways. ClueGo parameters were set as follows: Go Term Fusion selected; only display pathways with $p$ values ≤ 0.05; GO tree interval, all levels; GO term minimum genes, 3; threshold of 4% of genes per pathway; and a kappa score of 0.42. GO terms are presented as nodes and clustered together based on the similarity of genes present in each term or pathway. The most significant term was chosen as a representative of the group (Benjamini–Hochberg correction).

**Statistics**. A Wilcoxon matched-pairs signed-rank test was used, with subsequent Bonferroni correction to account for multiple testing, to evaluate whether there was a statistically significant difference in the level of one given marker at a given time point post-infection when compared to the baseline level (day 0; Figs. 1c, d and 3b, c; Supplementary Figs. 3a, b, e, f, 4a, b, e, f, 5d, and 7a–d, f–i). Baseline level in blood consisted in the median of 3–6 pre-infection values per animal. In pLNs, the baseline value consisted in the median of 1–2 pre-infection values per animal.

Cytotoxic activity of NK cells in various conditions was compared to the condition without peptide using the Wilcoxon matched-pairs signed-rank test within each species (AGM, cynMAC and rhMAC; Fig. 7d). The same way, pre- and post-infection cytotoxic activity was compared using a Wilcoxon matched-pairs signed-rank test in cynMAC and in AGM (the same animal providing both pre- and post-infection samples; Fig. 8a–d and Supplementary Fig. 9b, c). For rhMAC, a Kruskall-Wallis was used as animals were independent and as we further distinguished SIV controllers from viremic animals (Supplementary Fig. 9b).

Differences in viral replication level as quantified by SIV p27 according to the culture condition was determined with a Wilcoxon matched-pairs signed-rank test (Fig. 7e). For the analyses with the mutated SIV clones, we also considered a model where target cells were assumed to be at a constant level in the absence of NK$^+$ cells and to decline exponentially in the presence of NK cells due to the changes in the NK/CD4 T cell ratio.

In order to illustrate specific gene sets in the genome-wide RNAseq analysis, we draw heatmaps based on variance-stabilizing transformed counts (Fig 2a, b, c, d; Figs. 3a and 4d, e; Supplementary Figs. 5a–c and 6b, c, e, f; see also chapter above). Correlation analyses were performed using Spearman's coefficient (Figs. 3d, e and 4a; Supplementary Figs. 3c, d and 4c, d). These statistical analyses were performed with Prism (GraphPad, La Jolla, CA) and SigmaStat software (Systat).

**Reporting summary**. Further information on research design is available in the Nature Research Reporting Summary linked to this article.

## Data availability
Source data are provided with this paper. Deep sequencing results have been deposited in the Gene Expression Omnibus database; the accession number is GSE140600. The authors declare that all other data supporting the findings of this study are available within the article and its Supplementary Information files, or are available from the authors upon request. Source data are provided with this paper.

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

## Acknowledgements

We are grateful for the excellent help from the veterinarians and staff at the IDMIT Center (Benoit Delache, Jean-Marie Helies, Raphaël Ho Tsong Fang and Julie Morin). We thank M. Mietsch and M. Daskalaki for comprehensive veterinary support, and S. Heine, J. Hampe, and N. Leuchte for excellent technical assistance. We also thank all the people from the Illumina Hiseq 2500 platform at the Institut Pasteur and especially Caroline Proux and Jean-Yves Coppee. The Biomics Platform (C2RT, Institut Pasteur) is supported by France Génomique (ANR-10-INBS-09-09) and IBISA NH was supported by the VRI Labex, the Fondation Jacquelin Beytout and Institut Pasteur, and PR was recipient of a PhD fellowship from the University Paris Diderot, Sorbonne Paris Cité. C.P. was supported by a MSDAvenir Postdoctoral Fellowship Grant. FK was supported by the DFG (CRC 1279 and SPP 1923. We would like to acknowledge grant support from ANRS, the Fondation Jacqueline Beytout, and the Fondation Les Ailes to M.M.T., from the NIH (R01AI143457) to M.M.T. and R.K.R. and from MSDAvenir to A.S.C. and M.M.T.; J.L.H. and M.P. were supported by NIH grant R01AI116379 to M.P. and by ORIP/OD award P51OD011132 to Yerkes National Primate Research Center. J.N. was supported by the DFG (SPP 1937, SFB TR57), the Hector Foundation, and the DZIF (TTU-HIV). We gratefully acknowledge the support to IDMIT from the French government: Investments for the Future program for infrastructures (PIA) through the ANR-11-INBS-0008 grant as well as from the PIA grant ANR-10-EQPX-02-01 to the FlowCyTech facility at IDMIT. We equally acknowledge the Investments for Future grant ANR-10–INSB–04 to support the UtechS Photonic BioImaging (Imagopole) and C2RT facilities at Institut Pasteur.

## Author contributions

N.H. and M.M.T. designed the study; N.H. designed the experiments; N.H., P.R., C.Pe, R.G., and B.J. performed experiments; N.H., P.R., and J.N. designed the peptides; N.H., B.J., and U.S. quantified the viral loads; C.S. and F.K. constructed the infectious SIVagm clones; N.H. and E.B. performed the prediction analyses; C.S.H. and C.Pa provided samples; N.H., R.L., and H.V. performed the bioinformatic analyses; N.H., P.R., and Y.M. performed the statistical analyses; N.H., P.R., J.H., R.L.G., R.K.R., M.P., A.S.C., B.J., and M.M.T. analyzed the data; V.C., R.L.G., and B.J. coordinated the animal studies; M.M.T. obtained the funding; N.H. and M.M.T. wrote the manuscript and all co-authors reviewed it.

## Competing interests

The authors declare no competing interests.
