## [Peer Review File · Nature Communications]

Reviewers' Comments:

Reviewer #1:

Remarks to the Author:

In this revised version of their manuscript, the authors have done an excellent job responding to the comments from the reviewers. The point-by-point response is thoughtful and thorough. The additional data provided are interesting, and when the authors did not fully address specific points, the reasons are clearly explained.

This paper is a relevant contribution to the field. This reviewer has no more concern.

Reviewer #2:

Remarks to the Author:

I think that the authors have addressed the major concerns within the scope of what is feasible for a revision.

Reviewer #3:

Remarks to the Author:

In this resubmission, Huot et al. have substantially revised the original manuscript in response to reviewer comments. However, the manuscript has changed dramatically from the original submission to the point it should be considered as a new submission. Not every major concern from the original review was addressed appropriately. Instead a significant amount of new data has been added that does not necessarily support the major points of the manuscript. As presented, the manuscript is very complicated and challenging to follow. In particular, there is an overwhelming amount of data in the figures, making them difficult to interpret. Indeed, the major points the authors are attempting to convey are often lost as the reader attempts to wade through the myriad results presented in each figure. Instead of a focused insight into the function of MAC vs AGM NK cells, the main point of the paper is obscured in the complicated figures presented.

Major comments:

Considering the amount of data presented, a more comprehensive statistics analysis should be pursued examining multiple testing error or FDR.

Not all NK receptors have been shown to have an impact on recognition of SIV/HIV infected cells. The manuscript would benefit from a more focused and connected look at these receptors.

The paper suffers from a lack of in-depth or mechanistic explorations of its sequencing findings. A pertinent example is THEMIS which has been strictly associated with T cells and TCR signaling.

The author's do not establish a link between the first half (NK cell differentiation) and second half of the paper (MHC-E). Perhaps these findings would be best presented as two separate papers.

Point by point reply to reviewers' comments:

(reviewers' comments are indicated in *italics* and our replies in blue)

First of all, we would like to thank again the reviewers for their comments. These were very helpful for improving the clarity of our data and the manuscript.

REVIEWER COMMENTS**Reviewer #1**

In this revised version of their manuscript, the authors have done an excellent job responding to the comments from the reviewers. The point-by-point response is thoughtful and thorough. The additional data provided are interesting, and when the authors did not fully address specific points, the reasons are clearly explained.

This paper is a relevant contribution to the field. This reviewer has no more concern.

Reviewer #2:

I think that the authors have addressed the major concerns within the scope of what is feasible for a revision.

Reviewer #3

In this resubmission, Huot et al. have substantially revised the original manuscript in response to reviewer comments. However, the manuscript has changed dramatically from the original submission to the point it should be considered as a new submission. Not every major concern from the original review was addressed appropriately. Instead a significant amount of new data has been added that does not necessarily support the major points of the manuscript. As presented, the manuscript is very complicated and challenging to follow. In particular, there is an overwhelming amount of data in the figures, making them difficult to interpret. Indeed, the major points the authors are attempting to convey are often lost as the reader attempts to wade through the myriad results presented in each figure. Instead of a focused insight into the function of MAC vs AGM NK cells, the main point of the paper is obscured in the complicated figures presented.

Major comments:

Considering the amount of data presented, a more comprehensive statistics analysis should be pursued examining multiple testing error or FDR.

We totally agree that a comprehensive statistics analysis is essential and evaluation of FDR necessary for multiple comparisons. We apologize that the statistical assays we performed were not explained sufficiently clear. We have in the previous version used the Bonferroni correction to adjust for the false discovery rate. We consulted a biostatistician to verify all the

statistical assays of our study and to complete with additional tests whenever necessary in the revised manuscript. We have thus examined multiple testing error. We now describe in more detail in the method section all the statistical methods applied for each results. We also now explicitly mention for each figure and graph which statistical method was used. We hope that this clarifies the analyses in the manuscript and thank the reviewer for this thoughtful comment.

Not all NK receptors have been shown to have an impact on recognition of SIV/HIV infected cells. The manuscript would benefit from a more focused and connected look at these receptors.

The reviewer is obviously right. Not all NK receptors have an impact on recognition of SIV/HIV infected cells. Since NK cells in African green monkeys have been only poorly described so far, we indeed first performed a large analysis of the diversity of the NK cells in blood and the lymph nodes of AGMs to identify the NK cell subsets that expand upon SIVagm infection in lymph nodes (Figures 1-3). The data combined indicated a role of the NKG2 axis and therefore, the subsequent analyses (figures 4 and 5) were focused on the interaction between NKG2A and target cells. We added a few sentences to make this more clear.

The paper suffers from a lack of in-depth or mechanistic explorations of its sequencing findings. A pertinent example is THEMIS which has been strictly associated with T cells and TCR signaling.

We agree with the reviewer that we did not explore the function of THEMIS in our study. We acknowledge this now in the revised version (page 13). We have however performed mechanistic exploration of our findings. For instance, the genome-wide RNA sequencing revealed the presence of adaptive NK cells in the lymph node of SIVagm-infected AGM and we have set up assays to functionally demonstrate the NKG2-dependent adaptation of the NK cells in response to SIVagm infection in AGMs (Figure 5 f-i). In order to address the reviewer's comment, we removed THEMIS from the abstract, clarified the known role of Themis in the discussion and the lack of further mechanistical exploration of the signaling pathway in this study (page 13).

The author's do not establish a link between the first half (NK cell differentiation) and second half of the paper (MHC-E). Perhaps these findings would be best presented as two separate papers.

We apologize if the link was not sufficiently clear. We added a few sentences to better explain the link and improve the syntax throughout the manuscript for better explanation of our findings.